# A meta-analysis of the ecological and economic outcomes of mangrove restoration

Jie Su [1✉], Daniel A. Friess[2,3] & Alexandros Gasparatos [4,5]

Mangrove restoration has become a popular strategy to ensure the critical functions and economic benefits of this ecosystem. This study conducts a meta-analysis of the peer-reviewed literature on the outcomes of mangrove restoration. On aggregate, restored mangroves provide higher ecosystem functions than unvegetated tidal flats but lower than natural mangrove stands (respectively RR' = 0.43, 95%CIs = 0.23 to 0.63; RR' = −0.21, 95% CIs = −0.34 to −0.08), while they perform on par with naturally-regenerated mangroves and degraded mangroves. However, restoration outcomes vary widely between functions and comparative bases, and are mediated by factors such as restoration age, species, and restoration method. Furthermore, mangrove restoration offers positive benefit-cost ratios ranging from 10.50 to 6.83 under variable discount rates (−2% to 8%), suggesting that mangrove restoration is a cost-effective form of ecosystem management. Overall, the results suggest that mangrove restoration has substantial potential to contribute to multiple policy objectives related to biodiversity conservation, climate change mitigation and sustainable development.

[1] Graduate Program in Sustainability Science - Global Leadership Initiative (GPSS-GLI), Graduate School of Frontier Sciences, The University of Tokyo, Kashiwa City, Japan. [2] Department of Geography, National University of Singapore, Singapore, Singapore. [3] Centre for Nature-based Climate Solutions, National University of Singapore, Singapore, Singapore. [4] Institute for Future Initiatives (IFI), The University of Tokyo, Bunkyo-ku, Tokyo, Japan. [5] Institute for the Advanced Study of Sustainability (UNU-IAS), United Nations University, Shibuya-ku, Tokyo, Japan. ✉email: jie.su@s.k.u-tokyo.ac.jp

Mangroves are a highly biodiverse and productive ecosystem that provide multiple ecosystem services to local and global communities. Mangroves contribute to global climate regulation through carbon sequestration and storage[1], offer habitat and nursery grounds to many species[2], protect local communities from coastal hazards[3] and provide cultural ecosystem services such as ecotourism[4] and education. Although the ecological and socioeconomic importance of mangroves has been emphasised repeatedly in the past, mangroves were still lost globally between 2000 and 2016 at an average annual rate of 0.13%, with substantial inter-country variation[5]. Mangroves are expected to decline further due to various interconnected drivers including deforestation[6], pollution[7] and climate change[8]. Many of the world's remaining mangroves are degraded and fragmented[9].

Rapid mangrove loss, fragmentation and degradation create strong incentives for their restoration across their range to replace lost habitat and ecosystem services. There have been numerous efforts globally to conserve and restore mangroves; almost 2000 km² of mangroves have been planted over the past 40 years, with as much as 8000 km² of previously deforested mangrove areas remaining biophysically suitable for restoration[9]. Early mangrove plantation efforts were often geared towards silviculture, but the attention has increasingly shifted to restoring key ecological functions and socioeconomic benefits of mangroves[10]. To increase restoration success, many stakeholders now advocate for multi-species restoration and hydrological rehabilitation (e.g. the modification of water flow and drainage)[11,12], while other integrated engineering approaches are also being promoted[13].

An increasing number of field studies have assessed the ecological outcomes of mangrove restoration actions, such as the effect of mangrove restoration on ecological functions including carbon sequestration[14], nursery for fisheries[15] and heavy metal accumulation[16]. Similarly, studies have quantified the economic costs and benefits of mangrove restoration[17]. Results from these studies show divergent trends depending on their focus, context and methodology, and cannot be easily generalised as they typically focus on distinct contexts[18]. Previous reviews of the ecological and socioeconomic outcomes of mangrove restoration were mostly narrative syntheses of individual case studies from different parts of the world[10,19–21]. However, the quantitative synthesis of mangrove restoration outcomes can help identify research trends and gaps, as well as guide policy and action for biodiversity conservation and other relevant domains such as climate change mitigation.

Here we conduct a meta-analysis to quantify the outcomes of mangrove restoration and their relative magnitudes, as a means of providing a quantitative estimate of mangrove restoration performance. We use a combination of statistical tools to examine (a) restoration outcomes for a range of biogeochemical, ecological and other functions across different comparative bases (i.e. restoration mangroves vs. natural mangroves, naturally regenerated mangroves, degraded mangroves or unvegetated tidal flats), (b) the effect of diverse factors such as restoration age, approach, tree species and region, on restoration outcomes and (c) the economic costs and benefits of mangrove restoration. To ensure the proper assessment of restoration outcomes, our meta-analysis only includes observations that contained in the same peer-reviewed paper the performance of a restored mangrove for a given function(s), compared to one of four comparative bases (i.e. natural mangroves, naturally regenerated mangroves, degraded mangroves and unvegetated tidal flats) in the same area and environmental conditions (see Methods section). The meta-analysis reveals that mangrove restoration provides higher ecosystem benefits over unvegetated tidal flats, while generally lower than natural mangroves. However, restoration outcomes depend on restoration age, species and restoration method. We also find that mangrove restoration is cost-effective with positive benefit-cost ratios under variable discount rates. The results are synthesised to provide the evidence base for an ecological and economic case for mangrove restoration, and the possible implications for policy and research.

## Results

**General literature patterns**. Ecological outcomes were reported in 88.8% of reviewed studies ($N = 167$), economic costs/benefits in 14.9% of studies ($N = 28$) and both ecological and economic aspects in only 4.3% of studies ($N = 8$). The studies spanned a total of 22 countries and regions, mostly in East and Southeast Asia (China, 55 studies; Vietnam, 20 studies; Philippines, 17 studies) (Fig. 1). Mangroves were mostly restored with one species (175 cases), rather than two or more species (43 cases). The most commonly planted species were *Rhizophora apiculata* and *Rhizophora mucronata* (32 cases), *Avicennia marina* (31 cases) and *Kandelia obovata* (27 cases), though not all studies mentioned the target or actual restoration species (Supplementary Fig. 3). Studies have steadily increased over time (Supplementary Fig. 4).

**Ecological effects of restoration**. Reviewed cases spanned 26 individual ecosystem functions, with carbon sequestration and biomass production being the most prevalent among the identified studies (Supplementary Fig. 5 and Supplementary Tables 3 and 4). However, there were a lack of peer-reviewed studies reporting restoration outcomes at the same site and in the same study for paired restored mangroves and reference sites for some functions (e.g. photosynthetic performance, soil protozoa diversity, epibiotic diversity, control of invasive species and sediment accretion). For this reason, we eventually examined in more depth in the meta-analysis restoration outcomes for 21 out of 26 initially identified functions. This was conducted across four comparison groups, namely (a) restored mangroves vs. natural mangroves, (b) restored mangroves vs. unvegetated tidal flats, (c) restored mangroves vs. naturally regenerated mangroves and (d) restored mangroves vs. degraded mangroves (Fig. 2).

Figure 2a suggests that compared to natural mangroves, restored mangroves have on aggregate a lower ability to deliver different functions (RR′ = −0.21, 95% CIs = −0.34 to −0.08). Furthermore, when compared to natural mangroves, restored mangroves provide lower levels of biogeochemical functions (RR′ = −0.23, 95% CIs = −0.34 to −0.12) and relatively similar levels of ecological functions (RR′ = −0.16, 95% CIs = −0.46 to 0.14). Wave dissipation was the only identified function that does not fall under the biogeochemical or ecological function categories, with restored mangroves having a much lower capacity to dissipate waves compared with natural mangroves (RR′ = −0.69, 95% CIs = −1.05 to −0.34). Similarly the levels of most individual functions are much lower for restored mangroves compared to natural mangroves. For instance, restored mangroves have lower level of carbon sequestration (RR′ = −0.33, 95% CIs = −0.53 to −0.14) and nitrogen accumulation (RR′ = −0.39, 95% CIs = −0.67 to −0.11) than similar comparator natural mangroves. Conversely, the levels of GHG emissions (RR′ = 0.10, 95% CIs = −0.14 to 0.34) and phosphorus accumulation (RR′ = 0.10, 95% CIs = −0.21 to 0.17), and most of the individual ecological functions were not much different between restored and natural mangroves.

Figure 2b suggests that when compared to unvegetated tidal flats, restored mangroves have better restoration outcomes (RR′ = 0.43, 95% CIs = 0.23 to 0.63) and biogeochemical functions (RR′ = 0.46, 95% CIs = 0.24 to 0.68). Such patterns are also evident for individual functions such as carbon sequestration (RR′ = 0.64, 95% CIs = 0.35 to 0.94), nitrogen accumulation (RR′ = 0.45, 95% CI = 0.19–0.72), wastewater treatment (RR′ = 0.28, 95% CIs = 0.26 to 0.29) and wave

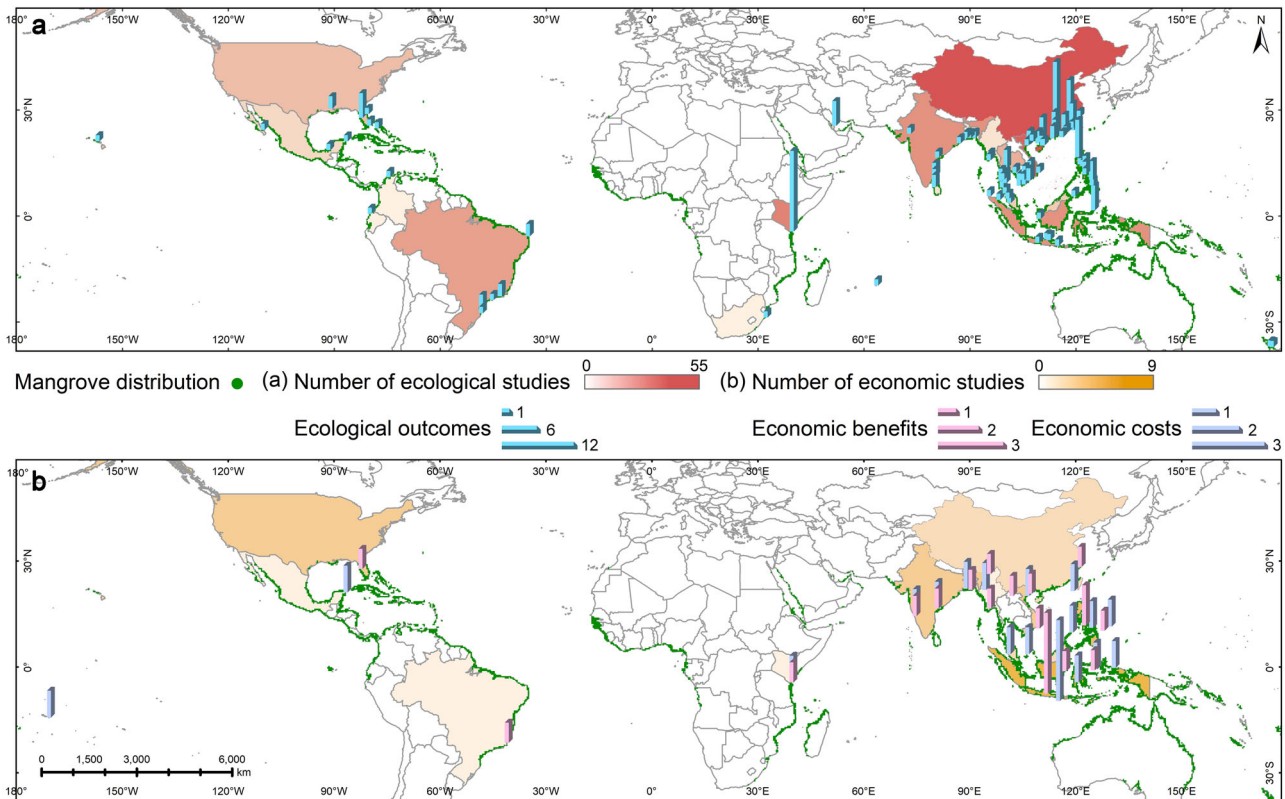

**Fig. 1 Distribution of reviewed studies.** Panel **a** presents the number of ecological studies, **b** number of economic studies. The bars in the map indicate the number of studies in specific locations (i.e. same coordinate). Source data are provided as a Source Data file. The global distribution of mangrove can be obtained at UNEP-WCMC with the identifier Giri et al.[98].

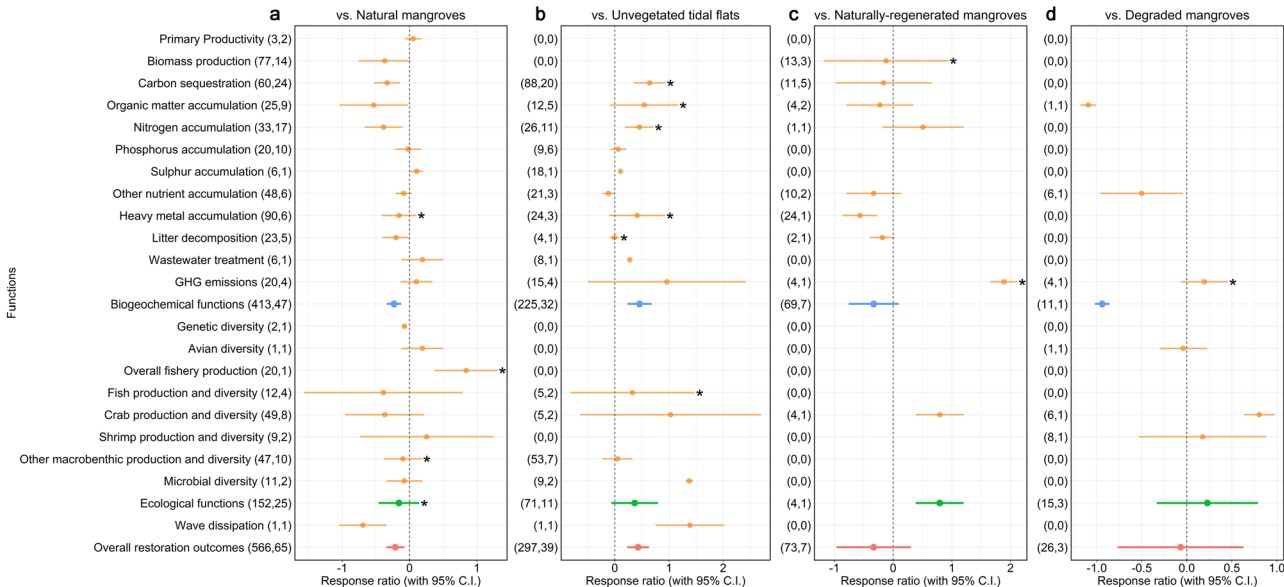

**Fig. 2 Mean effect size for different functions.** Panel **a** restored mangroves vs. natural mangroves, **b** restored mangroves vs. unvegetated tidal flats, **c** restored mangroves vs. naturally regenerated mangroves and **d** restored mangroves vs. degraded mangroves. Bars around the means denote 95% CIs. If a bar falls in the positive side and does not intersect with zero we interpret that the restored mangrove provides this specific function at a higher level than the comparative basis, and the opposite if it falls in the negative side of the forest plot. The first and second numbers in parentheses indicate, respectively, the number of observations and the number of studies included in each calculation. Bars with asterisks denote functions for which publication bias was detected. For the definitions of individual functions and aggregated categories refer to Supplementary Table 1. The overall restoration outcome for each comparative base was estimated as the pooled effect size of all individual functions. Source data are provided as a Source Data file.

dissipation (RR′ = 1.39, 95% CIs = 0.75 to 2.02). However, this is not evident for all individual functions, as for example there is no major difference for phosphorus accumulation (RR′ = 0.06, 95% CIs = −0.08 to 0.21), crab production and diversity (RR′ = 1.03, 95% CIs = −0.64 to 2.70), and fish production and diversity (RR′ = 0.33, 95% CIs = −0.82 to 1.84), among others.

There is a smaller volume of literature comparing the outcomes of restored mangroves with naturally regenerated or degraded mangroves in the same studies for the same site. Still, the meta-analysis indicates that restored mangroves have similar levels of restoration outcomes compared to naturally regenerated mangroves of the same age, and degraded mangroves (RR′ = −0.58, 95% CIs = −2.25 to 1.09, RR′ = 0.13, 95% CIs = −0.72 to 0.97, respectively) (Fig. 2c, d). When compared to naturally regenerated mangroves, only the level of heavy metal accumulation was lower in restored mangroves (RR′ = −0.57, 95% CIs = −0.87 to −0.27). There was no major difference for other functions such as carbon sequestration (RR′ = −0.16, 95% CIs = −0.98 to 0.66) and other nutrient accumulation (RR′ = −0.23, 95% CIs = −0.80 to 0.14). Conversely, restored mangroves exhibit higher levels of crab production and diversity compared to both naturally regenerated mangroves (RR′ = 0.79, 95% CIs = 0.38 to 1.20) and degraded mangroves (RR′ = 0.80, 95% CIs = 0.64 to 0.97). However, as the number of studies containing matched pairs of restored and naturally regenerated or degraded mangroves was quite limited, the outcomes of our meta-analysis for these comparisons should be interpreted with caution.

The Cochran's Q statistic test showed significant heterogeneity across the different restoration outcomes (Supplementary Table 5). As outlined in the next sections, subgroup analysis and meta-regression were conducted to identify the potential effect of different factors on the heterogeneity of the pooled effect sizes.

**Effect of stand age.** The ages of the studied stands ranged between 1 and 70 years old, with most studies reporting results from mangrove restoration projects between 1 and 15 years old (78.8% of cases) (Supplementary Fig. 6). The meta-regression using restoration age as a predictor suggests a significant correlation between overall restoration outcomes and stand age when the restored mangroves are compared to natural mangroves and unvegetated tidal flats (Supplementary Fig. 7). When looking into individual functions, functions such as biomass production and carbon sequestration significantly increased with mangrove age (estimated coefficient $\beta = 0.16$, $P < 0.0001$, $\beta = 0.03$, $P < 0.0001$, respectively) when comparing with natural mangroves. Similar increasing patterns are also observed for functions such as organic matter accumulation ($\beta = 0.15$, $P < 0.0001$) and crab production and diversity ($\beta = 0.24$, $P < 0.0001$), when comparing restored mangroves with unvegetated tidal flats. Conversely when comparing restored mangroves with natural mangroves, we observe a slight but statistically significant decrease for crab production and diversity ($\beta = −0.03$, $P = 0.0001$). This indicates that young restored mangroves may provide better nursery and habitat to larger crab populations than older restored mangroves, with crab production and diversity even in mature restored mangroves being lower compared to natural stands.

**Effect of restoration species.** The subgroup analysis did not suggest that any individual mangrove tree species has better restoration outcomes compared to other species in monospecific restoration settings (Fig. 3a) when comparing restored mangroves with paired natural mangroves. Monospecific restoration with popular mangrove tree species (i.e. *K. obovata*, *S. apetala*) generated considerable ecological effects compared to unvegetated tidal flats (Fig. 3a). The

restoration outcomes for individual functions varied between monospecific and mixed-species restored mangroves when compared to natural mangroves and unvegetated tidal flats (Supplementary Fig. 8). For example, when compared to natural mangroves, restored mangroves using mixed-species performed better than monospecific ones for biomass production (RR′ = −0.41, 95% CIs = −0.90 to 0.07 for monospecific vs. RR′ = −0.24, 95% CIs = −0.54 to 0.06 for mixed-species). However, for some ecosystem functions, monospecific restored mangroves generated higher outcomes when compared to natural mangroves (RR′ = 0.02, 95% CIs = −0.20 to 0.24 vs. RR′ = −0.33, 95% CIs = −0.61 to −0.06 for heavy metal accumulation) and unvegetated tidal flats (RR′ = 0.74, 95% CIs = 0.38 to 1.10 vs. RR′ = 0.51, 95% CIs = 0.13 to 0.88 for carbon sequestration) (Supplementary Fig. 8). Overall restored mangroves with mixed-species have better restoration outcomes when compared to naturally regenerated mangroves (RR′ = −0.44, 95% CIs = −0.88 to 0.00 vs. RR′ = −0.05, 95% CIs = −0.31 to 0.22) (Fig. 3a).

**Effect of restoration method.** In most of the reviewed cases restoration was performed through conventional planting (i.e. plantation) (96.2%), with only eleven articles reporting the outcomes of mangrove restoration through hydrological rehabilitation. The subgroup analysis suggests that plantation and hydrological rehabilitation have comparable overall restoration outcomes (Fig. 3b), but we need to note the much lower number of hydrological rehabilitation studies ($N = 5$) compared to studies on restoration through plantation ($N = 83$). For instance, compared to natural mangroves, the overall restoration outcomes of hydrological rehabilitation outperformed mangroves restored through planting (RR′ = −0.10, 95% CIs = −0.60 to 0.39 for hydrological rehabilitation vs. RR′ = −0.22, 95% CIs = −0.36 to −0.08 for plantation), while the results are inverse when compared to unvegetated tidal flats (RR′ = 0.28, 95% CIs = −0.51 to 1.06 vs. RR′ = 0.44, 95% CIs = 0.23 to 0.66).

**Effect of species origin.** In terms of overall restoration outcomes, the restoration projects that included the planting of exotic species (10% of observations) performed better than native species (90%) both when compared to unvegetated tidal flats (RR′ = 0.62, 95% CIs = 0.23 to 1.01 for using exotic species vs. RR′ = 0.38, 95% CIs = 0.16 to 0.59 using native species) and natural mangroves (RR′ = −0.21, 95% CIs = −0.39 to −0.02 vs. RR′ = −0.25, 95% CIs = −0.41 to −0.09) (Fig. 3c). However, due to limited observations it is not possible to estimate the effect size for individual functions for restoration projects that use exotic species compared with unvegetated tidal flats. When compared to natural mangroves, mangroves restored with exotic species perform on par with native species for some functions such as phosphorus accumulation (RR′ = −0.03, 95% CIs = −0.31 to 0.25 vs. RR′ = −0.02, 95% CIs = −0.29 to 0.26) but lower for other functions such as carbon sequestration (RR′ = −0.57, 95% CIs = −0.95 to −0.18 vs. RR′ = −0.32, 95% CIs = −0.54 to −0.09) (Supplementary Fig. 9).

**Regional variation.** Most of the studies assessing mangrove restoration outcomes focused in Southeast and East Asia, comprising 62.5% of the observations for comparisons with natural mangroves and 78.1% of the observations for comparisons with unvegetated tidal flats. The results of the subgroup analysis for comparisons between restored mangroves and natural mangroves do not indicate that restored mangroves perform better in any one specific region (Fig. 3d). However, when compared to unvegetated tidal flats, the overall restoration outcomes of restored mangroves were positive in East Asia (RR′ = 0.35, 95% CIs = 0.14 to 0.55), South America (RR′ = 1.89, 95% CIs = 1.46

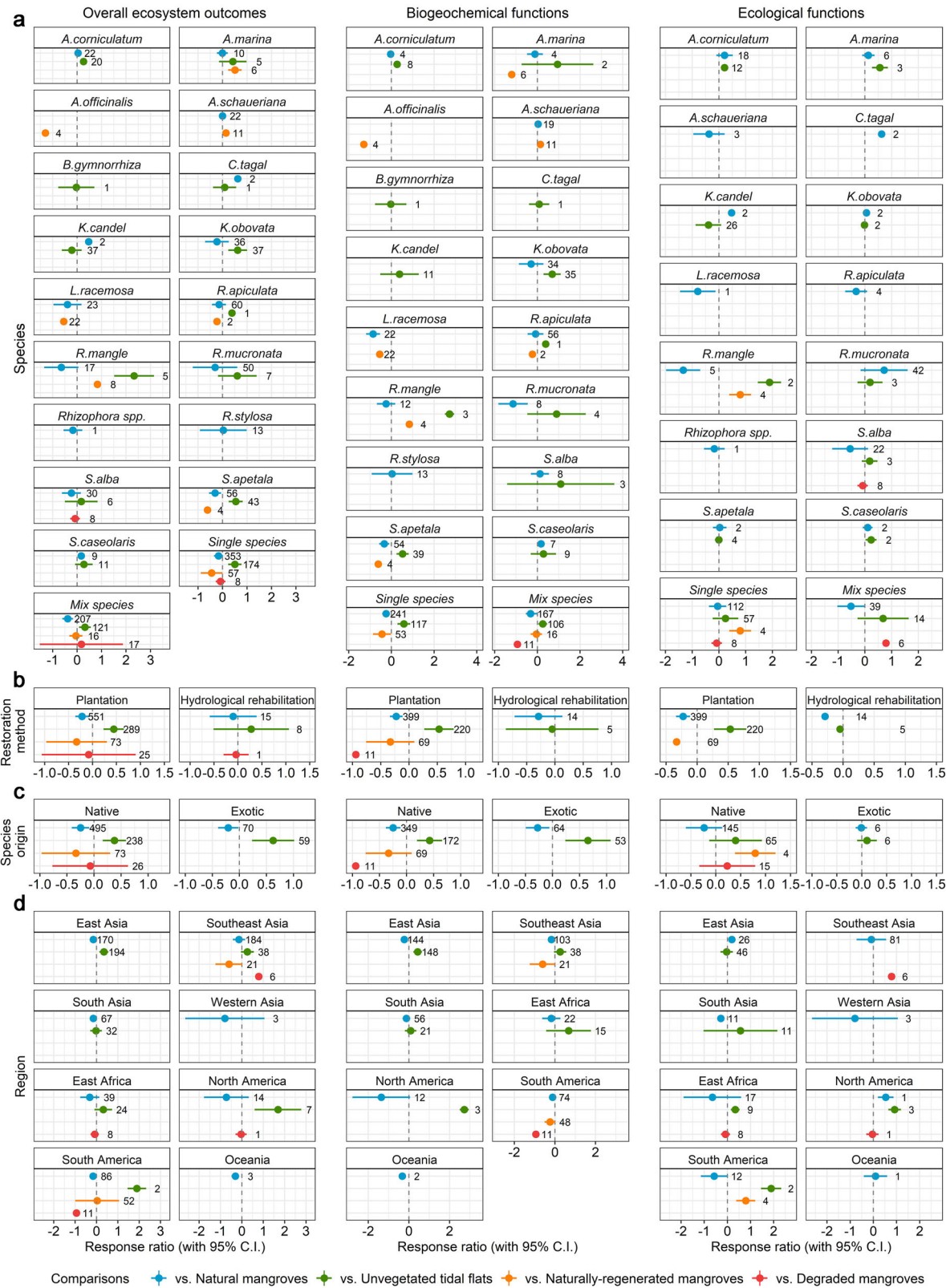

**Fig. 3 Subgroup analyses.** Panel **a** species, **b** restoration method, **c** species origin and **d** region. Bars around the means denote 95% CIs. The numbers at the right side of the confidence intervals denote the number of observations. If a bar falls in the positive side and does not intersect with zero we interpret that the restored mangrove provides this specific function at a higher level than the comparative basis, and the opposite if it falls in the negative side of the forest plot. Source data are provided as a Source Data file.

**Table 1 Economic benefits (in 2019 USD ha$^{-1}$ yr$^{-1}$) and costs (in 2019 USD ha$^{-1}$) from mangrove restoration.**

| | Restored mangroves | | | | | | Natural mangroves | | |
| --- | --- | --- | --- | --- | --- | --- | --- | --- | --- |
| | Obs. | Min. | Median | Mean | SE. mean | Max. | Obs. | Mean | SE. mean |
| Total value | 6 | 146.44 | 8713.62 | 19491.47 | 84048.87 | 510759.55 | – | – | – |
| Fisheries | 7 | 253.17 | 837.48 | 5291.81 | 3475.17 | 25593.05 | 20 | 10579.65 | 7352.79 |
| Timber production | 5 | 76.19 | 257.65 | 398.36 | 164.80 | 958.01 | 17 | 3500.39 | 2534.35 |
| Timber and honey | 1 | – | – | 6573.50 | – | – | – | – | – |
| Non-timber forest products | 1 | – | – | 87.62 | – | – | – | – | – |
| Climate regulation | 4 | 55.06 | 246.65 | 205.38 | 51.37 | 273.17 | 14 | 899.92 | 65189.63 |
| Coastal protection | 3 | 597.20 | 3420.82 | 2675.62 | 1052.98 | 4008.85 | 14 | 1296.46 | 355.02 |
| Wastewater treatment | 1 | – | – | 3566.41 | – | – | 7 | 3342.01 | 1937.89 |
| Recreation and tourism | 2 | 23.50 | 99.78 | 99.78 | 76.28 | 176.06 | 10 | 5468.47 | 4429.53 |
| Education and research | 1 | – | – | 1946.06 | – | – | 4 | 322.95 | 198.27 |
| Water | – | – | – | – | – | – | 2 | 2286.90 | 2059.42 |
| Regulation of water flows | – | – | – | – | – | – | 2 | 685.86 | 55.10 |
| Maintenance of generic diversity | – | – | – | – | – | – | 6 | 110.08 | 42.74 |
| Maintenance of life cycles of migratory species | – | – | – | – | – | – | 8 | 2352.44 | 1191.97 |
| Maintenance of soil fertility and nutrient cycling | – | – | – | – | – | – | 1 | 718.50 | – |
| Protection from erosion | – | – | – | – | – | – | 10 | 1133.95 | 381.76 |
| Aesthetic information | – | – | – | – | – | – | 1 | 561.32 | – |
| Sum | – | – | – | 20844.56 | – | – | – | 33258.8 | – |
| Total cost | 20 | 23.22 | 1269.24 | 209312.14 | 188367.79 | 3771126.76 | – | – | – |
| Plantation cost | 18 | 309.46 | 844.09 | 9627.69 | 6663.61 | 115766.57 | – | – | – |
| Maintenance cost | 7 | 45.73 | 73.47 | 6747.79 | 6629.68 | 46525.06 | – | – | – |
| Engineering cost | 10 | 200.70 | 19543.72 | 63757.07 | 31176.21 | 290807.29 | – | – | – |
| Labour cost | 6 | 4.10 | 617.08 | 23712.38 | 21712.68 | 132042.25 | – | – | – |
| Transportation cost | 6 | 6.24 | 26.90 | 24.63 | 3.86 | 33.99 | – | – | – |

The economic values of ecosystem services from natural mangroves were derived from a review of ecosystem service valuation studies from natural mangroves[23] after transforming them from USD ha$^{-1}$ yr$^{-1}$ of different years to 2019 USD ha$^{-1}$ yr$^{-1}$ using appropriate deflators (see Methods section). Crossbars indicate no applicable data. Source data are provided as a Source Data file.

to 2.32) and North America (RR′ = 1.69, 95% CIs = 0.60 to 2.78), while in South Asia there was no clear pattern (RR′ = −0.02, 95% CIs = −0.29 to 0.25).

**Sensitivity analysis and publication bias.** We identified several observations for which Cook's Distance was greater than traditional threshold of $4/n$ in some functions, suggesting considered high influence[22] (Supplementary Fig. 10). After excluding these outliers, we find that the effects of these outliers on the pooled effect size are rather minor with same or similar magnitude and direction of effect size and its 95% CIs, and thus the results are robust (Supplementary Table 6).

Temporal change tests show no significant correlation between the reported pooled effect size and publication year for overall restoration outcomes, biogeochemical functions, ecological functions and most individual functions across all group comparisons (Supplementary Fig. 11). Although the effect to some individual functions (e.g. organic matter accumulation in restored mangroves vs. unvegetated tidal flats) show significant correlation with publication year ($P = 0.005$), we can still infer that our results are robust as the estimates are small-scale ($\beta = −0.12$).

The funnel plot and Egger's test suggest that the results of the meta-analysis are robust without significant asymmetry, for the overall restoration outcomes, biogeochemical functions and ecological functions for all four comparisons. The only exception is for ecological functions for the comparison of restored vs natural mangroves. However, for some individual functions and comparisons (e.g. heavy metal accumulation and crab production and diversity in restored vs natural mangroves), the results of the Egger's test indicated the possibility of publication bias and/or

considerable heterogeneity across cases and/or studies, possibly due to unmeasured variables (Supplementary Fig. 12).

**Cost-benefit analysis.** We identified in the peer-reviewed literature nine types of economic benefits associated with mangrove restoration, which represent different ecosystem services (Table 1). Market price was the most commonly used method to elicit the economic value of these services, especially for provisioning services (e.g. timber production, fisheries) and climate regulation. Stated preference methods such as the contingent valuation method (CVM) and choice experiments have been used to elicit the total value of mangrove restoration, as well as of distinct ecosystem services such as recreation and tourism, and fisheries (Supplementary Fig. 13).

However, the actual estimates and number of studies varied substantially between types of benefits (Table 1), with few studies on cultural ecosystem services. Considering the lack of adequate samples and the variance in most of the original economic estimates, it was not possible to conduct a proper meta-analysis of the economic benefits, comparing observations of restored mangroves and other systems within the same studies and sites, as was done for ecological outcomes. However, to some degree it is appropriate to compare unpaired samples for economic benefits, because all values from different studies were adjusted from local currencies to 2019 USD, using appropriate conversion factors (see Methods section).

The simple comparison of our results with a review of ecosystem service valuation studies from natural mangroves[23] suggests a lower economic value for most of ecosystem services from restored mangroves (Table 1). This differential is high for

some services such as timber production, climate regulation and recreation and tourism (Table 1).

Mangrove restoration studies reported six main cost categories; plantation cost (including the cost of plants and planting), maintenance cost, engineering cost (e.g. pit digging, materials), labour cost, transportation cost and total cost (Table 1 and Fig. 4). Without considering extreme values, the range of total mangrove restoration costs was 23.22 to 371,326.75 USD ha$^{-1}$ with a median value of 1,097.16 USD ha$^{-1}$. The minimum total cost (23.22 USD ha$^{-1}$) was reported in a restoration project using the direct propagule dibbing method, which is different from conventional seedling transplantation[24].

The benefit-cost ratio of mangrove restoration ranged from 10.50 to 6.83 (for discount rates of −2–8%) when summing up the total benefits from each ecosystem service, and from 3.36 to 2.19 when using the mean value of the estimated total economic benefits (Supplementary Table 8). Both estimates suggest that mangrove restoration has considerable economic returns on investment. However, the benefit-cost ratio of natural mangroves

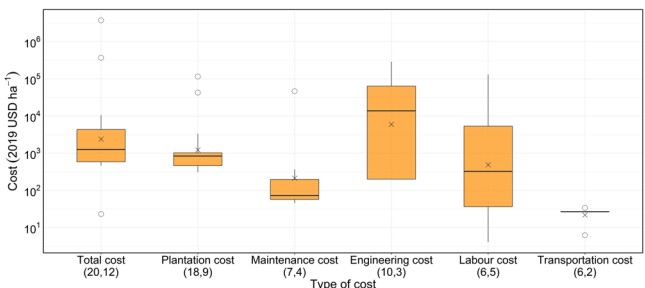

**Fig. 4 Range of mangrove restoration costs.** The first and second numbers in parentheses indicate, respectively, the number of observations and the number of studies for each cost category. The midlines in box-whisker plots represent the median values and the crosses (×) are the mean values. The bottom and top edges of each box indicate the 25th and 75th percentiles, respectively. The whiskers extend to 1.5 times the interquartile range and outliers are marked using dots. Source data are provided as a Source Data file.

can be as high as 16.75 under discount rates of −2%, which suggests that maintaining existing natural mangroves is more cost-effective than restoring degraded mangroves.

## Discussion

This meta-analysis suggests that there is an ecological case for mangrove restoration. However, the actual ecological outcomes of mangrove restoration (both in terms of magnitude and direction) vary substantially, depending on the type of function and comparative basis (Fig. 5). Unsurprisingly for most functions, restoration outcomes are mostly lower compared to natural mangroves (RR' = −0.21, 95% CIs = −0.34 to −0.08), but better compared to unvegetated tidal flats (RR' = 0.43, 95% CIs = 0.23 to 0.63). Restoration outcomes for many biogeochemical and ecological functions (e.g. biomass production) are influenced by stand age[25], limiting the ability of restored mangroves to provide such functions to the same level as mature natural stands. The overall outcomes of mangrove restoration perform on par with naturally regenerated mangroves (RR' = −0.58, 95% CIs = −2.25 to 1.09) or degraded mangroves (RR' = 0.13, 95% CIs = −0.72 to 0.97). This is consistent with previous meta-analyses of the restoration outcomes for wetland biodiversity and ecosystem services[26,27].

The effects outlined above bundle multiple individual functions with different levels of evidence. In this sense it is important to appreciate the restoration outcomes for individual functions. However, as discussed below, restoration outcomes for individual functions depend on the comparative base, as well as various other factors. This high variability needs to be considered when guiding restoration decision-making and action. For example, there was no difference between restored and natural mangroves for some individual ecological functions such as fish, crab and other macrobenthic fauna production and diversity. Furthermore, the levels of some biogeochemical functions, such as heavy metal accumulation, phosphorous accumulation and wastewater treatment, are comparable to natural mangroves. The above patterns reflect similar observations associated with the restoration of agro-ecosystems[28] and grasslands[29].

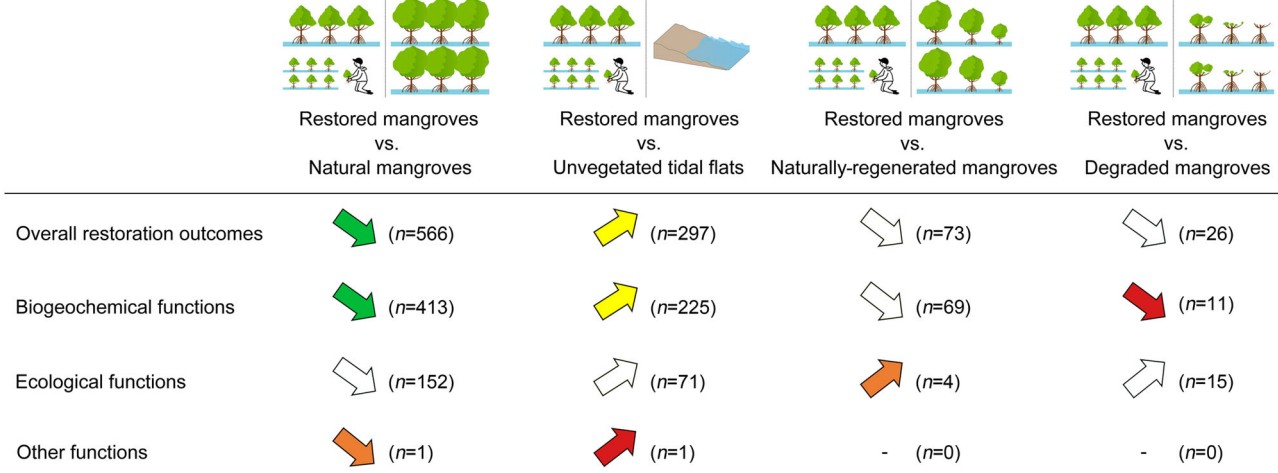

**Fig. 5 Summary of mangrove restoration outcomes.** All comparisons are between restored mangroves and the respective land use indicated at the top of the figure. Upward and downward arrows denote positive and negative effects respectively. Colours denote the interpretive benchmark of effect size values[99]. Green: small effects (absolute value of 0–0.30); yellow: medium effects (absolute value 0.30–0.60); orange: large effects (absolute value 0.60–0.90); red: very large effects (absolute value >0.90). For these coloured arrows the 95% CIs did not intersect with zero (see Methods section). Arrows in white indicate there is no major difference between restored mangroves and paired reference sites (i.e. 95% CIs intersected with zero). The numbers in parentheses indicate the number of observations (n). The crossbar (−) denotes that there were no applicable results for the specific function. Source data are provided as a Source Data file.

Individual studies identify numerous factors that affect the outcomes of mangrove restoration, such as vegetation characteristics (e.g. species, age, tree density, height, root)[30,31], sediments (e.g. depth, total N, total P)[32,33] and various environmental variables (e.g. location, salinity, debris)[34,35]. As a result, and given the context-specificity of these factors, the estimated outcomes of mangrove restoration are very heterogeneous. For this study, we delved deeper on the effect of some general factors on mangrove restoration, specifically restoration age, tree species (including monospecific or mix-species plantation), species origin (native or exotic), restoration approach and restoration sites (Fig. 3).

Restoration age has an effect on overall restoration outcome, and some individual ecological functions. For instance, the levels of individual functions such as biomass production, carbon sequestration, crab and fish production and diversity, are affected by tree age, as it is significantly correlated with the level of these functions (Supplementary Fig. 7). These findings are consistent with studies in individual sites that have focused on the influence of stand age on carbon accumulation[36,37] or changes in fish/crab assemblages[38,39]. For example, degraded and naturally regenerated mangroves require up to 40 years to reach the biomass levels and other structural characteristics of natural undisturbed mangroves[25,40]. However, there are some exceptions among individual functions that are not sensitive to stand age, such as GHG emissions, heavy metal accumulation, and shrimp production and diversity. These functions and processes may be related to other factors such as the time elapsed since disturbance, rather than stand age.

Mangrove restoration with multiple species is generally assumed to perform better than monospecific restoration, potentially because multiple species may lead to niche complementarity, while monospecific mangroves have a limited capacity to support benthic and terrestrial communities due to lower habitat complexity, which in turn underpins many other functions[10,41]. When compared to natural mangroves, our analysis reveals that restored mangroves with more diverse restoration species outperform monospecific plantation in terms of biomass production (Supplementary Fig. 8), which is consistent with a case study conducted in Hainan, China[42]. However, results also suggest that in some situations, monospecific restoration can perform better than mixed-species restoration for specific functions, when using natural mangroves and unvegetated tidal flats as the comparative basis (Supplementary Fig. 8). Individual studies also suggest that mixed-species restoration projects do not always have better outcomes than monospecific restoration projects, as for example shown in field experiments that found no significant difference in productivity between mixed-species and monospecific restoration with *A. marina*[43]. The benefits of monospecific restoration should be assessed carefully on a function-by-function basis, and only used where needed for a particular function (e.g. ecosystem design)[44]. Any potential benefits of monospecific restoration should be weighed against combinations of outcomes, or other outcomes of mangroves that were not measured in this study, such as biodiversity.

When it comes to the origin of restoration species, a recent meta-analysis of differences in the growth of native and exotic mangrove species in restored mangroves has implied that exotic mangrove species grow better than co-occurring native mangrove species in their introduced regions[45]. Similar findings appear in our meta-analysis that exotic mangrove species generate better overall restoration outcomes when compared to natural mangroves and unvegetated tidal flats (Fig. 3c). However, these results should be interpreted with caution, considering the relatively low number of samples for exotic species in comparison with unvegetated tidal flats. When compared to natural mangroves, our results imply that native species perform better than exotic

species for carbon sequestration (Supplementary Fig. 9), which is consistent with the results of a field study suggesting that native species monocultures (*K. obovata*) may have a higher carbon accumulation rate than exotic species (*S. apetala*) monocultures[46].

Hydrological rehabilitation without planting is an uncommon restoration method among the reviewed studies, but its effects for specific outcomes were on par with conventional planting methods[21] (Fig. 3b). However, a selection bias is apparent, as only sites that were successfully restored using conventional planting methods were reported in the literature, and thus included in this meta-analysis. More generally though, monospecific mangrove restoration through conventional planting methods show very low survival rates[47], particularly compared to hydrological rehabilitation or other methods where the original constraint on natural regeneration is addressed[13].

The substantial asymmetry found in some individual functions in funnel plots indicates publication bias for some specific restoration outcomes. Failure to publish results of low or no statistical significance or small effect size could be one of the reasons leading to publication bias[48]. This could be detected, for example, in funnel plots for carbon sequestration, organic matter accumulation, and heavy metal accumulation for restored mangroves compared to unvegetated tidal flats (see the significant right-skewed funnel plot in Supplementary Fig. 12). Such publication biases need to be taken into consideration when interpreting some of the results of this study. There is no single step to fully mitigate or account for publication bias in meta-analyses, as this is a feature of the underlying literature[49]. It is the responsibility of the academic community, including researchers, reviewers and editors, to ensure an unbiased literature[50].

The results also suggest that there is an economic case for mangrove restoration. The total economic benefits from restored mangroves amount to about two thirds of the benefits from natural mangroves (Table 1). The estimates of mangrove restoration benefits ranged widely (146.44-510,759.55 USD ha$^{-1}$ yr$^{-1}$), reflecting the significant differences in valuation methods, mangrove productivity, valuated ecosystem services, and wealth disparities between regions[51]. The relatively low economic benefits accruing from mangrove restoration might be due to the immaturity and lower diversity of restored mangroves, as in all economic studies the reported age of the restored trees was below 20 years (Supplementary Table 7), despite studies arguing that mangrove restoration may yield optimal levels of mean annual increment after 40 years[52]. Additionally, younger and monospecific mangroves are less productive compared to mature and diverse mangrove systems, which could have in turn affected ecosystem services related to fish productivity or timber production[53]. Unfortunately, we are unable to account for the variations among tree species, ages, and restoration methods due to the lack of sufficient information (Supplementary Table 7).

Depending on the study, the total costs of mangrove restoration range between 23.22 and 371,326.75 USD ha$^{-1}$, with a median value of 1,097.16 USD ha$^{-1}$. Even though the costs of the planting phase are relatively similar across studies and countries (low variance in Fig. 4), labour costs vary widely, potentially due to the large variations in employment costs and economic conditions between countries. In any case, total restoration costs fall within the reported estimates in other restoration contexts, e.g. coastal restoration (median of 2508 USD ha$^{-1}$)[54] and wetland restoration (69-828,033 USD ha$^{-1}$; median 4368 USD ha$^{-1}$)[55].

Overall, the cost-benefit analysis suggests a sound economic return on investment for mangrove restoration, even under high discount rates (range 10.50 to 6.83 under discount rate of −2–8%) (Supplementary Table 8). The sensitivity analysis shows comparable and higher benefit-cost ratios compared to similar

restoration contexts such as coastal wetlands (0.61–6.68)[56]. However, the delivery of some provisioning ecosystem services (e.g. fisheries, timber production) and cultural ecosystem services (e.g. recreation and tourism) may be prohibitively low for several years following restoration due to strict management and monitoring[57]. As discussed below, this might affect benefit-cost ratios, at least in the short-term, and should be factored in restoration decision-making, especially in relation to the opportunity cost of alternative land use options[58].

The findings of this study have major policy implications, considering that ecosystem restoration gains significant traction in global policy agendas, as the United Nations have declared 2021–2030 as the UN Decade on Ecosystem Restoration. In fact the restoration of coastal ecosystems such as mangroves have become integral in this debate[59]. Additionally, the concept of the Blue Economy that values the ecosystem services provided by coastal ecosystems has also gained international momentum[60]. It has been argued that mangrove restoration can contribute to transitioning to a Blue Economy in tropical regions[61], considering the significant ecosystem services provided by mangroves, including the sequestration of blue carbon[25].

When looking at our results at a highly aggregated level, there is a clear ecological and economic case for mangrove restoration. Restored mangroves provide many functions at higher levels than unvegetated tidal flats (which may include bare flats caused by the historical loss of mangroves) (Fig. 5), while for some critical functions such as carbon sequestration they can be on par with naturally regenerated mangroves (Fig. 2). In this sense, restored mangroves can be a cost-effective carbon sink, considering that mangroves are among the most carbon-rich forests, have a large GHG mitigation potential, and thus a possible role in climate change mitigation strategies[1,62]. This points to the potential of mangrove restoration in some countries to contribute meaningfully to the upcoming post-2020 Global Biodiversity Framework[63], the Paris Agreement[64] and the UN Sustainable Development Goal 14[65] (as well as other interlinked SDGs[66]).

That said, however, the levels of most individual functions are generally lower for restored mangroves compared to both natural and naturally regenerated mangroves (Figs. 2 and 5). This possibly applies also for returns-on-investment, though the benefit-cost ratio of mangrove conservation was not estimated in this study. Thus there is a clear need to continue prioritising natural mangrove conservation considering their extensive historical and ongoing global loss[9].

Throughout this study we observed large variability, uncertainty and lack of information for some mangrove restoration outcomes, as well as the possible actions that can improve its performance and leverage its potential for biodiversity conservation, climate change adaptation/mitigation and overall sustainability. We need to bridge these knowledge gaps to ensure that a robust evidence base guides relevant restoration actions around the world. First, we need to understand better the restoration outcomes for the understudied individual functions and economic effects. For example, we could find only a few studies on restoration outcomes related to bird populations and shoreline protection (Fig. 2). The latter is particularly surprising, considering that shoreline protection has been a key driver of mangrove restoration efforts in several countries after natural disasters such as the Indian Ocean tsunami and Typhoon Haiyan[67]. Similarly, very few economic studies have focused on the cultural ecosystem services provided by restored mangroves (Table 1), in part because many cultural ecosystem services (such as spiritual and religious values, education and aesthetic appeal) are intangible and difficult to monetise. At the same time, more studies should attempt to provide robust paired analysis for

functions in restored vs. degraded mangroves, which were particularly limited in our meta-analysis (Fig. 2).

Ideally, such ecological and economic knowledge should be used to design appropriate evidence-based restoration plans, policies and payment systems[68,69]. Compared to the extensive (yet fragmented and incomplete) literature on restoration outcomes for different functions (Fig. 2), there is less published information on attempts to mobilise such evidence for the design and implementation of conservation mechanisms such as Payments for Ecosystem Services (PES) schemes, to both motivate local community engagement in mangrove restoration and formulate proper compensation mechanisms for landowners[70]. Furthermore, we need to understand better the climate change mitigation potential of mangrove restoration vis-à-vis other mitigation options to, among others, identify funding options for mangrove restoration that are competitive with other climate change mitigation options.

Finally, when viewing mangrove restoration as a potential strategy to foster blue economic transitions, we need to note that even though restored mangroves provide substantial ecosystem services, the maximum provision of these services is not automatic and might require many years to reach appreciable levels[40]. This means that mangrove restoration might have important short-term opportunity costs to local communities. For example, substantial revenue can be generated through the conversion of mangrove forests for aquaculture uses[71]. In this sense, mangrove restoration actions might have negative economic outcomes to local communities, which might in turn reduce the desirability and acceptability of restored mangroves, or increase its vulnerability to further degradation. If mangrove restoration actions are to become an important driver towards a blue economy, then appropriate tools should seek to reduce such opportunity costs. Some promising options include blue carbon credits[72] or combinations of restoration with sustainable aquaculture models (e.g. integrated mangrove-aquaculture systems[73]). Such options would most likely be context-specific but their selection should be informed through evidence-based research.

## Methods

**Definitions of comparison groups.** Four comparisons were conducted in this meta-analysis (a) restored mangroves vs. natural mangroves, (b) restored mangroves vs. unvegetated tidal flats, (c) restored mangroves vs. naturally regenerated mangroves and (d) restored mangroves vs. degraded mangroves.

Restored mangroves encompass various interventions including restoration, rehabilitation, reforestation, regeneration and plantation. Other terms such as afforestation are often used interchangeably with restoration, even though this process refers to planting mangroves in areas that were previously not mangrove. This study uses restoration as an umbrella term that encompasses a range of human-driven activities involving the (re)generation of mangrove vegetation either in areas that were previously mangrove forest but lost or degraded, or new areas that are biophysically suitable for mangrove colonisation or afforestation. While we used the term restoration to cover all these interventions where mangrove cover has been increased, we do not make any judgement of the suitability of these activities.

Natural mangroves are generally undisturbed mangroves, i.e. mangroves that have not been affected by severe disturbances, either related to human activity (e.g. land use change, overexploitation) or natural processes (e.g. hypersalinization). Naturally regenerated mangroves are those that have been restored naturally or "passively" through natural seedling recruitment. Degraded mangroves refer to mangroves disturbed from natural processes or human activities (see above for examples). Unvegetated tidal flats include unvegetated or ephemerally vegetated mud and sand flats or abandoned aquaculture ponds, often found seaward of the mangrove forest in the lower intertidal zone.

**Literature selection.** We systematically searched the peer-reviewed literature to identify quantitative studies about the effect of mangrove restoration on different functions, as well as the economic costs and benefits of mangrove restoration. We focused on literature about active mangrove restoration rather than passive restoration (i.e. natural regeneration without human intervention). Articles were identified through ISI Web of Science Core Collection and Elsevier Scopus through

two iterative literature searches outlined below. We used the five selection criteria below for our study.

Criterion 1: for the restoration outcomes we consider quantitative assessments. We focus on empirical field studies and exclude studies from experiments in laboratories, microcosms, tanks, greenhouses or pots. The restoration age of the mangrove should be clearly stated, with newly planted seedlings (<3 months) not taken into consideration.

Criterion 2: for the meta-analysis, we focus on four comparison pairs: (a) restored mangroves vs. natural mangroves, (b) restored mangroves vs, unvegetated tidal flats, (c) restored mangroves vs. naturally regenerated mangroves, and (d) restored mangroves vs. degraded mangroves. The observations used in the meta-analysis should include mean, standard variation (standard deviation or standard error) and sample size for both land uses (i.e. restored mangroves vs. one of the respective comparisons mentioned above). To ensure the proper assessment of restoration outcomes we select observations that contained in the same paper/study the performance of a restored mangrove for a given function(s), compared to one of four comparative bases (i.e. natural mangroves, unvegetated tidal flats, naturally regenerated mangroves and degraded mangroves in the same area and environmental conditions.

Criterion 3: For the economic analysis, we select only studies specifically mentioning the monetary value of ecosystem services and the valuation method.

Criterion 4: We only estimate the effects of active restoration (i.e. plantation, hydrological rehabilitation). Passive restoration without human intervention (i.e. natural succession) is regarded as a naturally regenerated mangrove for comparison purposes (see Criterion 2).

Criterion 5: We do not consider studies assessing the ecological impacts of processes that interrupted mangrove restoration (i.e. impacts of tropical typhoons on restored mangroves).

We followed PRISMA protocol for study selection and inclusion[74] (Supplementary Fig. 1). The first search was performed on 22 December 2019 with no restriction on publication year using generic wording for restoration impact. We used the keywords "Mangrove* AND (restor* OR replant* OR rehabilitat* OR reforest* OR afforest* OR plant* OR recover* OR regener*) AND (assess* OR impact* OR outcome* OR effect* OR economic* OR valu* OR monetary OR ecosystem services OR valuat* OR benefit OR cost)". In total 3445 articles were identified, of which 2834 were discarded after a first screening of the title and abstract. During the second screening the full-text was read to judge its suitability for inclusion. Overall 151 articles representing 293 cases of mangrove restoration met the inclusion criteria.

The second search was conducted on 30 June 2020 using a much more refined keyword selection after obtaining a good understanding of the different function categories following the first search. We used the keywords "Mangrove* AND (restor* OR replant* OR rehabilitat* OR reforest* OR afforest* OR plant* OR recover* OR regener*) AND (biomass OR carbon OR nitrogen OR phosphorus OR crab* OR GHG OR fish* OR metal OR litter OR sediment OR sulphur OR gene* OR crustacea OR primary produc* OR photosynthe* OR organic OR wave OR storm OR shrimp OR microbial OR wastewater OR inva* OR control OR avian OR bird OR protozoa OR reptile OR mammal OR molluscs OR invertebrates)". In total 37 additional articles focusing on different functions were found.

Peer-reviewed datasets from published reviews were also taken into consideration, but economic/cost valuations from the grey literature and documents not in English were not covered in this study. In the end, a total of 188 articles with 395 cases were used in this study. A case refers to one tree species with specific restoration age and same or similar environmental conditions, even when there are two or more measurements of restoration outcome. The full list of all 188 reviewed articles considered in this systematic review is included at the end of the Supplementary Information.

**Critical appraisal of studies**. There is no standard methodology to assess the quality of the studies included in systematic reviews and meta-analyses[75]. In this study we adopted the checklist proposed for assessing the strength of evidence of ecosystem services and biodiversity conservation studies, which covers aspects of data collection, analysis, and reporting[76] (Supplementary Table 2). We modified and categorised the identified studies based on the quality of their evidence into those that have very strong evidence (score: >75%), strong evidence (score: 50-74%), moderate evidence (score: 25-49%), and weak evidence (score: <24%) (see Supplementary Table 2 for criteria and estimation).

As the systematic review sought to highlight the research landscape we sought to include the widest possible range of empirical studies, only excluding those with weak evidence, i.e. without underlying data[76]. However, for the meta-analysis, we only included studies with very strong or strong evidence, as a means of ensuring the high quality of the results.

Overall, the quality appraisal indicates that all studies included in our systematic review meet at least the basic quality requirement (i.e. moderate quality evidence). The quality scores of all studies (188) range between 40 and 100%, with a mean value of 86% (SD = 11%). Only two of the studies were categorised as containing moderate quality evidence (Supplementary Fig. 2), while 88% of studies were categorised as containing very strong quality evidence. For the studies included in the meta-analysis, the quality is even higher, with 86 of the 88 included

studies categorised as containing very strong quality evidence, and the rest strong quality evidence.

It should be noted that the quality of the studies focusing on the economic aspects of mangrove restoration is relatively uneven, when compared to the studies of ecological outcomes (Supplementary Fig. 2). Generally, the studies focusing on the ecological outcomes of mangrove restoration tend to adopt standardised protocols for sampling, data collection, and statistical analysis. Conversely, the studies focusing on the economic outcomes of mangrove restoration can range from those using robust experimental designs and methods (e.g. choice experiments, contingent valuation method) to those reporting restoration costs through simple narratives.

**Meta-analysis of restoration outcomes**. From each study we extracted quantitative data about the outcomes of mangrove restoration across different functions, as well as key restoration characteristics (e.g. age, species, restoration method, region). A total of 167 studies reported restoration outcomes about 26 types of functions using different variables (Supplementary Table 1). These 26 types of functions were categorised into three aggregate categories, namely biogeochemical functions, ecological functions, and other functions[77] (see Supplementary Table 1 for the definition of different individual functions). Although the three categories are overlapping between all[77], we classified the individual functions based on their dominant function (Supplementary Table 1). The estimations of restoration outcomes were conducted across four different types of paired comparisons, namely (a) restored mangroves vs. natural mangroves, (b) restored mangroves vs. unvegetated tidal flats, (c) restored mangroves vs. naturally regenerated mangroves and (d) restored mangroves vs. degraded mangroves. As mentioned above, the type of comparative basis was clearly indicated in the individual study, and observations were only included when they were paired both at the same sites and in the same study. In total 88 studies (representing 962 observations of different variables) met our criteria for inclusion in the meta-analysis.

To quantify the effect size of restored mangroves and reference systems within the same study we calculated the natural logarithm of response ratio (lnRR) (a unit-free index) as a measure of effect size of each observation. This approach has been widely applied in ecological meta-analyses[78,79]. The lnRR was calculated as ln ($\bar{x}$ restored mangroves /$\bar{x}$ natural mangroves or unvegetated tidal flats or naturally regenerated mangroves or degraded mangroves), where $\bar{x}$ is the mean value of a quantified variable. For those response variables predicted to correlate negatively with ecosystem function (i.e. emission of $CO_2$, $CH_4$ and $N_2O$), we inverted the sign of the lnRR for those "negative" variables before combining them with other functions to calculate the overall effect size. We extracted mean, statistical variation (i.e. standard error, standard deviation) and sample size for restored and reference groups for each variable.

Conventional models for meta-analyses (i.e. fixed-effect model and random-effect model) assume independence between the observed effects or outcomes obtained from a set of studies. However, non-independence is ubiquitous in ecological meta-analyses[80], as there are often multiple variables estimated in individual studies (e.g. Mg, Ca, and K accumulation when estimating other nutrient accumulation in the same study). As a consequence non-independence within individual studies is inevitable and considerable in our database. Thus we calculated the weighted effect size (RR') using multivariate models for the overall restoration outcomes and each type of function to model the clustering (and hence non-independence) induced by a multilevel structure in the data, where the non-independence in the observed/true effects or outcomes was accounted for.

We also illustrated the 95% confidence intervals (CIs) in forest plots. We interpreted that mangrove restoration has a clear positive or negative effect for a given function(s) and comparative base, if the 95% CIs does not intersect with zero in the forest plots. Thus if it falls on the positive side of the forest plot it means that the restored mangrove provides a given function(s) at a higher level than the comparative basis, and the opposite if it falls in the negative side of the forest plot. At this point we need to point that some studies have criticised the misuse of P-values or CIs as a dichotomous description of "statistical significance"[81,82]. Here we emphasise that the 95% CIs does not reflect the conventional statistical significance but manifests the range of probable effect size estimates with 95% confidence[83]. Thus in the narrative of the Results when using 95% CIs we avoid using the term "significant" to denote statistical significance in the traditional sense. Instead we interpret that if the 95% CIs of the restoration outcome for a given function(s) does not intersect with zero, then restored mangroves provide that given function(s) in a clear-cut higher (if the 95% CIs are at the positive side) or lower level (if the 95% CIs are at the negative side) than the comparative basis.

We used the Cochran's Q statistic (Qt) based on the $\chi^2$ test to test whether effect sizes were homogeneous across studies[84]. A significant Qt indicates that the variance among effect sizes is greater than that expected by sampling error alone.

Subgroup analyses were conducted for (a) restoration tree species (e.g. R. apiculata; R. mucronata; A. marina; single species; mix-species), (b) restoration methods (plantation; hydrological rehabilitation without planting), (c) restoration species origin (native; exotic) and (d) broad restoration region (e.g. Southeast Asia; East Asia; South America). These subgroup analyses were used to identify factors causing heterogeneity across studies, by inspecting the effect sizes and their differences within each subgroup. Despite the similarities between meta-regression

and subgroup analysis, we conducted meta-regression as it allows the use of continuous data (i.e. restoration age) as a predictor, and to check whether restoration age is associated with effect size differences[85]. Therefore, we extracted the information of study site (i.e. country, coordinates of study area), restoration method, tree species, restoration age and restoration species origin for each case.

**Sensitivity analysis**. We diagnosed outlier and influential observations using Cook's distance[86], which can be interpreted as the Mahalanobis distance between the entire set of predicted values once with the $i$th observation included and once with the $i$th observation excluded during model fitting. A possible outlier of observation was detected when a Cook's D of more than $4/n$ where $n$ is the number of observations for estimated functions/outcomes[22]. After detecting the significant outliers, the pooled effect sizes were recalculated by excluding the outlier observations to test the effect of the outliers on the results. We also used meta-regression to examine the relationship between effect size and publication year as the temporal change test of our results.

**Publication bias**. We applied the funnel plot and Egger's regression[87] to test for publication bias in all estimates, including overall restoration outcomes, and the aggregate and individual functions. The adopted procedure of above meta-analyses followed the guidelines proposed for biological meta-analyses[88], and were conducted using the "metafor"[89] package in R version 3.6.1.

**Cost-benefit analysis**. The inclusion criteria for the cost-benefit analysis were met by 31 observations spanning 10 types of economic benefits (including total value), and 67 observations spanning six types of restoration costs. All published values were standardised in 2019 USD ha$^{-1}$ yr$^{-1}$ following the procedure described in detail in the TEEB database[90]. In particular, the official exchange rates were used to convert the economic estimates reported in the original studies into the official local currency[91]. These values were then adjusted to 2019 local currency values using official gross domestic product (GDP) deflators. Finally, the purchase power parity (PPP) conversion factors were used to convert the value to 2019 USD.

The economic benefits of restoration were estimated for a period of 20 years at social discount rates of −2%, 4.5% and 8%, with the negative rate reflecting the possibility that conditions will deteriorate in the future due to ecological degradation and resource depletion[56]. The net-present annual total economic benefit per hectare is estimated starting in the 5th year through both summing up the estimated values for each ecosystem services and the mean value of the estimated total benefit by excluding the maximum value. To avoid double-counting, the net-present total cost (in 2019 USD ha$^{-1}$) was averaged for all total cost data, excluding the maximum and minimum value and with annual management cost component of up 2.5% of the capital cost[56]. The benefit-cost ratio (BCR) of mangrove restoration was then calculated for 20 years following Eq. 1:

$$BCR = \frac{|PV[Benefits]|}{|PV[Costs]|} = \frac{\sum_{t=5}^{T} B_{total}/(1+r)^t}{C_{total} + \sum_{t=1}^{T} C_{management}/(1+r)^t} \quad (1)$$

where $PV$ is the present value; $t$ is the year of calculation, $B$ is the total economic benefits; $C$ is the costs (including total initial cost and management cost); and $r$ is the discount rate.

**Limitations**. First, this study relied only on peer-reviewed literature. However, a large volume of information on tropical ecosystems can be found in the non-peer-reviewed literature[92], including mangrove restoration projects. Thus, this meta-analysis might not have included some relevant studies, and especially those reporting restoration project costs that are more commonly described in consultancy reports, reports to funders and other grey literature. Meanwhile, we believe that such omission may be less prevalent for studies on the ecological effects of mangrove restoration, as such information is normally collected in scientific research projects. In order to contain data of sufficient observations and replication for meta-analyses such project documentation would usually go through the peer review publication process. Despite this omission of non peer-reviewed studies it has been suggested that there is no difference when directly comparing effect sizes between published and unpublished studies in ecological meta-analyses[93].

Secondly, this meta-analysis was based on peer-reviewed papers extracted from databases containing literature predominantly published in the English language. However, studies on mangrove restoration may also be published in languages other than English such as Bahasa, Mandarin or Portuguese. English language publications have also been the basis of other meta-analyses on global coastal habitat restoration[94], but it may introduce biases, particularly in terms of the magnitude of mean effect sizes[95].

Third, none of the publication bias methods currently available has desirable statistical properties under extreme heterogeneity in true effect size[96]. The funnel plot and the Egger's test are both popular methods for testing whether small-study effects are present in a meta-analysis. However, publication bias is one of the causes of small-study effect and other factors can cause the skewed funnel plot such as the

choice of effect measures and chance. Thus, the funnel-plot method has been found inappropriate for heterogeneous meta-analyses[97].

**Reporting summary**. Further information on research design is available in the Nature Research Reporting Summary linked to this article.

## Data availability

The data that support the findings of this study are available in Figshare with the identifier https://doi.org/10.6084/m9.figshare.12901382.v6. The source data for plotting figures and tables can also be archived in the above link, except for Supplementary Tables 5, 6 and 8 and Supplementary Fig. 12, which are directly created using R (R-3.6.1) functions. The study quality assessment table is also available in the above link. The global distribution of mangrove can be obtained at UNEP-WCMC with the identifier https://doi.org/10.34892/1411-w728.

## Code availability

The R (R-3.6.1) code used in this study for meta-analysis is available in Figshare with the identifier https://doi.org/10.6084/m9.figshare.12901382.v6.

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

## Acknowledgements

This work is financially supported by Ministry of Education, Culture, Sports, Science and Technology, Japan (193156), and Japan Society for the Promotion of Science (Kakenhi B 19H04323). The authors thank Quanli Wang and Bingchao Yin for their valuable suggestions on improving Figures.

## Author contributions

All authors contributed intellectual input and assistance to this study. J.S. and A.G. designed the research. J.S. conducted literature search, data extraction and analysis. J.S. wrote the first draft of the manuscript, and D.A.F and A.G. contributed substantially to revisions.

## Competing interests

The authors declare no competing interests.
