## [Peer Review File · Nature Communications]

Reviewers' Comments:

Reviewer #1:

Remarks to the Author:

This manuscript addressed an important issue in mangrove ecosystems with the most biodiverse while with heavily degraded all over the world. A lot of efforts globally were made to conserve and restore mangroves during the past 40 years. This manuscript aiming to estimate the outcomes across different functions and economic costs of the natural and restored mangrove ecosystems. A meta-analysis of the peer-reviewed literature was conducted for the estimate. The studies spanned a total of 22 countries and regions, mostly in East and Southeast Asia (China, 51 studies; Vietnam, 23 studies; Philippines, 18 studies). These regions are facing the challenges of mangrove degradation while made great efforts in mangrove restoration. Thus, this study might potential to contribute to multiple policy objectives related to biodiversity conservation, climate change mitigation and sustainable development.

The followings are some comments to the authors for the revision of manuscript.

1. Since there were different types of mangrove restoration performance in different studies, how to group them as restored or natural. Especially those naturally-regenerated mangroves and restored, and the degraded mangroves? Please state in the methods.
2. Although there were some grey literature which were not covered in this study. The amount was big. Did you think the results would be drawn to another direction if included them? In the discussion, some about this might be helpful.
3. Why the functions of Ecological Function did not included the biodiversity of waterfowl? The waterfowl diversity was usually used for the evaluation of coastal wetland functions. The unvegetated tidal flats in mangrove ecosystems are important feeding habitats for waterfowl. In this function considered in the analysis, whether it would dropped the results in Figure 1 to another way?
4. Line 457, the sample size for restored and reference groups for each variable were not shown in Supplementary Table 2. Did you mean Supplementary Table 3.
5. In the reviewed cases spanned 26 functions, although carbon storage and biomass production were the most prevalent. How about the contribution of restored mangroves to the Blue Economic?

Reviewer #2:

Remarks to the Author:

Review of NCOMMS-20-35903-T

The authors present several meta-analyses of effect sizes that characterise differences between types of mangroves, for a range of 'ecosystem functions'. Given the importance of mangroves to biodiversity and ecosystem service provision, this is an important question. However, I have major concerns with the quality of the meta-analysis, detailed below.

As written, the manuscript is hard to follow in places. For example, in the abstract, I don't follow these sentences: "restoration outcomes depend between functions and comparative bases", and "restoration offers positive benefit-cost ratios across various discount rates". The manuscript would benefit from some heavy editing. I provide some suggestions below, which I hope are useful.

Here are some useful resources that will help in a future revision of this work:

- 1) Koricheva et al. 2013 - Handbook of Meta-analysis in Ecology and Evolution <https://www.jstor.org/stable/j.ctt24hq6n> I believe this book is currently free to download due to the pandemic
- 2) Systematic review protocol: Guidelines and Standards for Evidence Synthesis in Environmental Management: <https://www.environmentalevidence.org/information-for-authors>
- 3) Nakagawa et al. 2017 - Meta-analysis: Meta-evaluation of meta-analysis: ten appraisal questions for biologists <https://bmcbiol.biomedcentral.com/articles/10.1186/s12915-017-0357-7>

Major concerns with the meta-analytic method

- 1) Literature search – from the description, the literature search does not appear to be systematic nor transparent. In terms of the search terms, the taxa specified seem arbitrary: the authors included avian OR bird OR protozoa OR crustacean OR shrimp. What about mammals, inverts, reptiles? Why not fishes or molluscs? In the introduction, the authors discuss the importance of mangroves as a habitat for fishes, yet this service is not reflected in the search.
- 2) The methods are vague as written and do not inspire a lot of confidence in scientific rigour. Little detail is given on the screening and eligibility, or the data extraction and coding. Insufficient detail has been given about the systematic review methodology used to obtain and critically appraise the studies.
- 3) Question and effect size – the set-up of the analysis, concerning the choice of reference and restored sites is not justified or well-described. The choice of effect size metric is not explained – why Hedges' g ? But also $\ln R$? Also, some of the ecological functions have positive benefits for society – e.g. biomass production, NPP, while GHG emissions have negative benefits. Therefore a positive and negative effect size for these different functions would have positive and negative implications for society, so it doesn't make sense to pool them. Many ecological functions are not explained. What is 'microbial community'? How is this a function? How is it measured? What is 'macrobenthic production'? Perhaps the authors could add a table with explanations of the functions.
- 4) Sample sizes – they are very low for some of the 'ecological functions': e.g. I see 'avian community' has just one effect size, from just one study, and 'fishery production' has 21 effect sizes from one study. This is not representative?

- 5) Non-independence – multiple effect sizes were obtained from each study, and the authors do not describe if/how this was addressed in their meta-analysis. This can lead to erroneous conclusions.
- 6) Meaningful synthesis? The authors present an overall ‘ecological function’ effect size across all, very different, ecological functions. This is of questionable utility.

More specific comments

Introduction

General: It is a little clunky, and feels like it doesn't quite flow. There are quite a few vague statements that could be better articulated/quantified. I hope my comments below help.

L50 – ref?

L50 – remove ‘Among others,’

L54 – ‘emphasized repeatedly’ – by who? ‘repeatedly’ begs >1 ref, but 1 ref is given.

L55 – what is the unit of loss? Land area? I'm struggling to get a sense of the true extent of loss, as only a rate between 2000-2012 is given. Anything more recent? What total extent remains/has been lost?

L59: Remove ‘The’

L60: ‘As a result’ – delete. It's not necessary, as you set it up in the previous sentence

L61: ‘many efforts’ – sounds vague.

L66: what's hydrological rehabilitation?

L67: ‘Other integrated approaches are increasingly being promoted’. This is vague. Not sure what point is being made here. The ends of the paragraphs should lead the reader on to the next. I'm at the end of the second paragraph now, but still don't get a sense of what the big question is, what's important?

L71: ‘nursery habitat’ – that's not a ‘process’. Change to ‘habitat provision’?

L72: here you state that ‘**Similarly**, studies have quantified the economic costs and benefits’, but in the first sentence of this paragraph you have already stated that ‘field studies have assessed the ecological and/or economic outcomes...’

L80: “adopts a meta-analysis approach” → uses meta-analysis? It's a method not an approach?

L80: “assess” → quantify

L81: long sentence, can be cut down considerably by just saying what you quantified

L82: ‘various’ – avoid this word. It sounds vague. A range of?

Methods

L419: I think you should have searched for mangrove*, rather than mangrove, crab* rather than crab. There are several examples of this.

L438: “valuations from the “grey literature” and documents not in English were not covered in this study”. Why not? How much was there? How do you think this impacted your results?

L439: 'In the end'. After what? There has been no description of a robust systematic review protocol, e.g. inclusion/exclusion criteria. Was there any critical appraisal?

L446: "about 26 types of functions using different variables". Vague. How were these classified?

L447: In total 89 studies (representing 975 observations of different functions) met our criteria for inclusion in the meta-analysis. What criteria? Why is this in your section 'Meta-analysis of restoration outcomes'. It should go in literature selection.

L449: "namely". Unnecessary – you name three?

L450-452: This is too late to be describing the comparisons you set out to make. They need to be clearly justified.

L463: 'which provides conventional statistical significance, but also information that cannot be obtained from p values' I don't follow. What 'information'?

L465: So you also calculated lnR as well as Hedges' g? How did you get to the mean percentage of change from lnR? Why did you calculate both effect sizes?

L477: species – not everyone will know what taxa 'Rhizophora apiculata' is. Make clear it's the tree species.

L447: plantation; habitat rehabilitation – describe the difference. Where does natural regen fit? You use that term in your intro.

L479: 'origin' sounds more like the restoration method. I think you mean 'nativeness'. How did you classify whether species were native or not? What resource was used?

L485: 'Elements of the adopted methodology' – vague. What elements? What is the point of this sentence? Perhaps a bit army-wavey. In general, avoid single-sentence paragraphs.

L488: 'To test the robustness of the results' → vague. Robustness in relation to what? Influential studies?

L497: 'met in 30 cases'. What is a 'case' here? An article? A study? A comparison? Please define.

Results

'General literature patterns:' is a little vague for a subheading. Perhaps subheadings could be short statements about the results. Same for other subheadings.

L92 '88.8% of reviewed studies' say how many first. Also, 'reviewed' is vague – do you mean those that satisfied review criteria?

L102: '26 functions,' - we need a lot more explanation here.

Reviewer #3:

Remarks to the Author:

Su et al. conducted a meta-analysis on ecological and economic effects of mangrove restoration. The topic is clearly important and timely. As my expertise is in meta-analysis rather than mangroves, I largely focussed my review on the methods used in this meta-analysis. My specific comments are listed below.

1. Methods, line 315. Please specify which databases were used in Web of Science, which is not a database itself but a platform which allows access to a range of databases, and that range depends on institutional subscription. Therefore, your search is not repeatable unless you specify which databases within WoS (e.g. Core Collection) you had access to.
2. Please use past tense throughout the methods section, currently it is a mixture of past and present (e.g. line 459)
3. You say that you used standardized mean difference (Hedges' d) as a measure of effect size, but then on lines 465-466 you say that you calculated mean percentage of change from response ratios. Why then not use response ratios as your effect size measure, i.e. why use two different effect sizes?
4. Lines 476-483. You say that you conducted subgroup analyses to check for effects of restoration age, but then you also say that you conducted meta-regression to look at restoration age effects. Why do both analyses? Using restoration age as a continuous variable in meta-regression is more powerful approach statistically than comparing effects between studies categorized into different restoration age classes.
5. I was not clear how or whether in your analysis you took into account non-independence between multiple observations from the same study. For instance, presumably in many studies effects of restoration on particular process/function were assessed in multiple years so that multiple effect sizes for that response variable could be derived from the same study. Such repeated measures from the same site/study are not statistically independent and you need to control for this in the analysis, e.g. by including study as a random factor in your model. I could not find any mention in the methods of how or whether you dealt with this problem.
6. Supplementary Table 2. You categorized restoration outcomes into broad function type categories (biogeochemical, ecological, anthropocentric). Depending on the specific function measured, the direction/sign of the effect could be interpreted differently. For instance, for biogeochemical functions, higher biomass and carbon storage in restored mangroves is good, but higher emissions of GHG is bad. Similarly, for ecological functions, higher avian diversity in restored mangroves is good, but higher number of invaded species is bad. Did you invert the sign of the effect size for such 'negative' functions before combining them with other functions to calculate the overall effect size? This needs to be explained.
7. As part of sensitivity analysis you have performed leave-one-out meta-analysis (Supplementary figure 16). It is not clear from this figure and figure caption what is Y axis on all the graphs and what red, blue and black line represent. Instead of leave-one-out analysis, I would prefer to see analysis of effects of outliers (studies with particularly large negative or positive effects) on the results.
8. Publication bias and interpretation of trim and fill results. You state (lines 209-210) that results of trim and fill analysis indicate that publication bias might exist. This is incorrect. Trim and fill procedure does not test whether publication bias exists or not, rather, it tests how much your results would change IF publication bias exists. Results of trim and fill procedure that you present in Supplementary Table 6 worry me a great deal because they show that your main results would be reversed if publication bias exists as the direction and/or statistical significance of your main effects changes dramatically when you adjust for possible publication bias. Given that many of your funnel plots are very asymmetrical and suggest that publication bias is likely, it basically suggests that the main results that you are reporting are not robust to publication bias and likely to be overturned if grey literature or unpublished studies were included in the analysis. This is a big concern and needs to be discussed more thoroughly.
9. Geographic bias. One important role of meta-analysis is to highlight current gaps in our knowledge on the topic. Looking at Supplementary figure 3 and comparing mangrove distribution with locations of studies included in your meta-analysis, it is clear that there is a strong geographic bias with lots of studies from Asia but e.g. no studies from Australia. This needs to be acknowledged and discussed in your paper. Is this because mangroves in Australia have not been degraded as much as in Asia and America and hence restoration efforts have not been as

widespread? Is there a danger of extrapolating results from your analysis based largely on studies from Asia and America to other regions? Have you tried to account for regional differences in your analysis?

10. You have found evidence of increase in magnitude of the effect size with publication year in some categories of studies (Supplementary figure 17) with reversal in the direction of the effect occurring in a matter of decade. This is very striking, yet you do not really discuss any potential causes of this increase in the ms. Could it be an artefact of more recent studies being conducted in different regions or using different restoration methods?

11. Main results. You state in the abstract: "We find that restored mangroves provide many individual functions at levels higher or on par with natural mangroves, naturally-regenerated mangroves, degraded mangroves and unvegetated tidal flats", but Figure 1 tells a very different story. You acknowledge later in the abstract that restoration outcomes vary depending on measured functions and comparative bases, but this needs to be explained more carefully, e.g. for most biogeochemical functions restored mangroves perform much worse than natural mangroves or naturally regenerated mangroves.

Response to Reviewers

Reviewer #1 (Remarks to the Author):

This manuscript addressed an important issue in mangrove ecosystems with the most biodiverse while with heavily degraded all over the world. A lot of efforts globally were made to conserve and restore mangroves during the past 40 years. This manuscript aiming to estimate the outcomes across different functions and economic costs of the natural and restored mangrove ecosystems. A meta-analysis of the peer-reviewed literature was conducted for the estimate. The studies spanned a total of 22 countries and regions, mostly in East and Southeast Asia (China, 51 studies; Vietnam, 23 studies; Philippines, 18 studies). These regions are facing the challenges of mangrove degradation while made great efforts in mangrove restoration. Thus, this study might potential to contribute to multiple policy objectives related to biodiversity conservation, climate change mitigation and sustainable development.

The followings are some comments to the authors for the revision of manuscript.

Thanks for the feedback. We revise following your comments.

Since there were different types of mangrove restoration performance in different studies, how to group them as restored or natural. Especially those naturally-regenerated mangroves and restored, and the degraded mangroves? Please state in the methods.

All types of mangrove considered in this study (i.e. restored, natural, naturally-regenerated or degraded) were clearly indicated as such in the original individual studies we used to extract the data for this meta-analysis. So this is not a value judgment of the authors of this study, but clearly stated information in each of the source studies.

For the benefit of the reader we now also add one paragraph in the Methods section to clarify the definition of natural mangroves, naturally-regenerated mangroves, degraded mangroves and unvegetated tidal flat. We also add a sentence that the status of the mangrove was clearly stated in each of the source studies.

Although there were some grey literature which were not covered in this study. The amount was big. Did you think the results would be drawn to another direction if included them? In the discussion, some about this might be helpful.

The decision to exclude grey literature rests on our concerns over possible quality issues as the overwhelming majority of these studies are not peer-reviewed, and report the outcomes of restoration projects for which the reporting organizations have received the funding to undertake the restoration projects.

In our opinion the exclusion of grey literature will not affect the meta-analysis of ecological outcomes, as few such reports delve deep in the assessment of the ecological outcomes of restoration using the types of paired comparisons that are used in our meta-analysis. A more significant fraction of reports might report the economic costs of mangrove restoration projects.

The exclusion of grey literature is rather common in meta-analyses such as the one reported in this study. The exclusion of grey literature was already illustrated and justified on the first paragraph of the Limitations. We believe that as this is a methodological issue it is more logical

to be covered in the Methods rather than the Discussion.

We should also note that some previous studies have tested and suggested that there is no major difference in effect sizes reported in published and unpublished studies in ecological meta-analysis (Møller et al. 2005).

Why the functions of Ecological Function did not included the biodiversity of waterfowl? The waterfowl diversity was usually used for the evaluation of coastal wetland functions. The unvegetated tidal flats in mangrove ecosystems are important feeding habitats for waterfowl. In this function considered in the analysis, whether it would dropped the results in Figure 1 to another way?

As pointed by the Reviewer, waterfowl is important for unvegetated tidal flats and tidal marshes, where open space is required for foraging. While some birds can be found in the mangrove, bird diversity is generally more important for tidal flats. In the first round of literature review, we used general words to identify the effects of mangrove restoration and we found few studies mentioning waterfowl.

Following this comment we search the databases (WoS and Scopus) using the keywords “Mangrove* AND (restor* OR replant* OR rehabilitat* OR reforest* OR afforest* OR plant* OR recover* OR regener*) AND waterfowl”, and still did not find a record regarding waterfowls that meet the selection criteria for this meta-analysis.

Line 457, the sample size for restored and reference groups for each variable were not shown in Supplementary Table 2. Did you mean Supplementary Table 3.

The Supplementary Table 2 (previous version, now is Supplementary Table 1) introduces the variables. We now delete this information to avoid confusion.

In the reviewed cases spanned 26 functions, although carbon storage and biomass production were the most prevalent. How about the contribution of restored mangroves to the Blue Economic?

We are not sure we understand the comment well, so apologies if we do not answer properly.

The blue economy is not an ecological function of mangroves or a distinct economic cost/benefit, so it cannot be included as an analytical category in the ecological or economic analysis.

The Blue Economy is an economic system that values and internalizes the benefits provided by healthy ecosystems such as mangroves. We now reflect better in a few sentences the possible contribution of mangrove restoration to the Blue Economy due to the provision of different ecological functions and the relatively high benefit-cost ratios across different discount rates.

We will be happy to revise further if the Reviewer wishes to clarify better this specific point.

Reviewer #2

The authors present several meta-analyses of effect sizes that characterise differences between types of mangroves, for a range of ‘ecosystem functions’. Given the importance of mangroves to biodiversity and ecosystem service provision, this is an important question. However, I have major concerns with the quality of the meta-analysis, detailed below.

As written, the manuscript is hard to follow in places. For example, in the abstract, I don’t follow these sentences: “restoration outcomes depend between functions and comparative bases”, and “restoration offers positive benefit-cost ratios across various discount rates”. The manuscript would benefit from some heavy editing. I provide some suggestions below, which I hope are useful.

Here are some useful resources that will help in a future revision of this work:

Koricheva et al. 2013 - Handbook of Meta-analysis in Ecology and Evolution <https://www.jstor.org/stable/j.ctt24hq6n> I believe this book is currently free to download due to the pandemic

- 1) Systematic review protocol: Guidelines and Standards for Evidence Synthesis in Environmental Management: <https://www.environmentalevidence.org/information-for-authors>
- 2) Nakagawa et al. 2017 - Meta-analysis: Meta-evaluation of meta-analysis: ten appraisal questions for biologists <https://bmcbiol.biomedcentral.com/articles/10.1186/s12915-017-0357-7>

Thanks for the productive feedback. In reality we have followed the PRISMA protocol for conducting systematic reviews and meta-analyses, but we agree that some points were not explicitly stated. We revise the abstract and main manuscript following all your recommendations. We hope our approach is much clearer now.

Major concerns with the meta-analytic method

Literature search – from the description, the literature search does not appear to be systematic nor transparent. In terms of the search terms, the taxa specified seem arbitrary: the authors included avian OR bird OR protozoa OR crustacean OR shrimp. What about mammals, inverts, reptiles? Why not fishes or molluscs? In the introduction, the authors discuss the importance of mangroves as a habitat for fishes, yet this service is not reflected in the search.

The keyword “fish” was included as a research keyword in the original analysis (it was the 7th keyword of the second string of keywords). Molluscs fell under the category of macrobenthic fauna. During the first literature search, we did not identify relevant papers containing information about the other indicated keywords (i.e., mammals, reptiles), so we had omitted them in the second search.

Following the comments of the Reviewer, in this revised version we expanded the keywords. In particular we added the keywords “mammals”, “invertebrates”, “reptiles” and “molluscs”, but we could not identify any new article meeting the inclusion criteria clearly stated in Supplementary Table 1 in first draft of Supplementary Information (now in main text following next comments).

The methods are vague as written and do not inspire a lot of confidence in scientific rigour. Little detail is given on the screening and eligibility, or the data extraction and coding. Insufficient detail has been given about the systematic review methodology used to obtain and

critically appraise the studies.

We had provided extensive information about the methodology in Supplementary Table 1 (First draft of Supplementary Information) with some summary version in the main text about the selection criteria for the studies, variables and scanning of studies. We are not sure if the Reviewer did not see this information or wants further information.

In summary, as stated above, we followed the good practices for systematic reviews and meta-analyses proposed by PRISMA, with a full flow diagram included as Supplementary Figure 2 (First draft of Supplementary Information, now is Supplementary Figure 1). We illustrated the data extraction process in the paragraph about response ratio and sub-group analysis.

We had created a repository entry in Figshare containing all datasets (and codes) to be made freely available for the benefit of the reader, as we believe it will be a very useful resource. However we did not make it public waiting to hear the outcome of the review. In retrospect we should have submitted it as supplementary files to facilitate the review process.

When it comes to the quality appraisal of the studies we had loosely followed some of the criteria suggested by Mupepele et al (2016) about assessing the evidence of individual studies on ecosystem services and biodiversity. However, it was not formalised.

Following the comments above, and in order to provide more information about our methodology:

- we add the study selection criteria in the main text rather than in the Supplementary Information;
- we define better the comparative bases in main text (following also a comment from Reviewer 1);
- we upload the database used in the meta-analysis (essentially showing the coding) and the code as supporting files for the benefit of the reviewer (these will be made freely available following publication);
- define better the analytical variables with citations in Supplementary Table 1 (Revised version) (see comment below)
- add additional information in Methodology about data extraction from the individual studies;
- formalise the quality appraisal of the individual studies following the criteria of Mupepele et al (2016), and include the outcomes of this assessment as a supporting file for the benefit of the reviewer.

We hope that collectively all this information inspires more confidence on the rigor of our study. If needed we will be happy to provide further information or expand the existing content subject to further suggestions from the Reviewer.

Question and effect size – the set-up of the analysis, concerning the choice of reference and restored sites is not justified or well-described. The choice of effect size metric is not explained – why Hedges' g ? But also $\ln R$?

Also, some of the ecological functions have positive benefits for society – e.g. biomass production, NPP, while GHG emissions have negative benefits. Therefore a positive and negative effect size for these different functions would have positive and negative implications for society, so it doesn't make sense to pool them.

Many ecological functions are not explained. What is ‘microbial community’? How is this a function? How is it measured? What is ‘macrobenthic production’? Perhaps the authors could add a table with explanations of the functions.

In the original manuscript, we used Hedge’s g as the effect size metric, which has been widely used in ecological meta-analyses. The mean response ratio ($\ln RR'$) was also calculated in order to estimate the mean percentage of change through $(e^{RR'} - 1) \times 100$. However, this information was not correctly indicated and calculated in the first version of manuscript.

In the revised manuscript, we recalculate the effect size using Response Ratio instead of Hedges’ d to avoid confusion, as this is also suggested by Reviewer 3. The use of response ratio is quite justified considering its use in nearly half of the meta-analyses in the field of plant ecology (Koricheva et al 2014). We add sentences justifying this methodological selection in the Methods.

The Reviewer is correct about the “negative” effects, and we appreciate for pointing it out. Following a similar comment by Reviewer 3 we now invert the sign of response ratio for these variables, before combining them to calculate the overall effect size. We indicate this in 1-2 sentences in Methods.

Regarding the last comment, we now add an extra column in Supplementary Table 1 in Supplementary Information (revised version) explaining each function, and providing a relevant citation. We hope that this makes much clearer to readers what each function refers to.

Sample sizes – they are very low for some of the ‘ecological functions’: e.g. I see ‘avian community’ has just one effect size, from just one study, and ‘fishery production’ has 21 effect sizes from one study. This is not representative?

The Reviewer is correct. Unfortunately, for some functions there is a general lack of studies, especially for some such as “restored mangroves vs naturally-regenerated mangroves” and “restored mangrove vs degraded mangroves”. This is a feature of the literature that is beyond the control of the authors.

We have pointed in appropriate parts of the manuscript that there should be caution when interpreting the results of the function with relatively low sample size and/or based on only one study. Furthermore, we point out in the Discussion that more studies should attempt to provide robust paired analysis for these functions and/or specific comparisons.

In any case, we believe that this is an interesting reflection/discussion point in its own right indicating major knowledge gaps that become particularly pertinent to be addressed, now that we enter the Decade of Restoration as designated by United Nations.

Non-independence – multiple effect sizes were obtained from each study, and the authors do not describe if/how this was addressed in their meta-analysis. This can lead to erroneous conclusions.

The Reviewer is correct that since there are more than one observation captured in some studies, the issue of non-independence might be considerable in our dataset. In the first draft of the manuscript indeed we did not consider this, which possibly led to inaccurate results and erroneous conclusions.

Following a similar comment by Reviewer 3, in this revised manuscript we use multivariate

models instead of standard meta-analytical models (fixed-effect model or random-effect model) to calculate the pooled effect sizes, so that the non-independence in the observed/true effects or outcomes is accounted.

Overall this change in analytical procedure corrects the results generated through the conventional meta-analytical models (random-effect model) used in the first draft. This revision increases the robustness of the meta-analysis as the results of the publication bias tests and the sensitivity analyses indicate that the results of the multivariate models are much more robust compared to the random-effect model used in the previous version.

Meaningful synthesis? The authors present an overall ‘ecological function’ effect size across all, very different, ecological functions. This is of questionable utility.

We see the reviewer’s point, and we considered this issue during the initial development of the manuscript. The point raised by the Reviewer is similar to any discussion over a composite index that aggregates multiple indicators. We believe their “utility” depends ultimately on the reader, and we cannot determine this a priori. Some readers might find it useful and others not.

We believe that what is more important is to point the large heterogeneity between functions and what this “aggregation implies”. We add some acknowledgements in the Discussion about the care that needs to be taken when interpreting the aggregated results.

More specific comments

Introduction

General: It is a little clunky, and feels like it doesn’t quite flow. There are quite a few vague statements that could be better articulated/quantified. I hope my comments below help.

We revise the introduction taking on board all comments below. We hope that this streamlines the introduction.

L50 – ref?

The reference is added.

L50 – remove ‘Among others,’

The statement is removed.

L54 – ‘emphasized repeatedly’ – by who? ‘repeatedly’ begs >1 ref, but 1 ref is given.

Additional references are added.

L55 – what is the unit of loss? Land area? I’m struggling to get a sense of the true extent of loss, as only a rate between 2000-2012 is given. Anything more recent? What total extent remains/has been lost?

We searched again and confirmed that this article and information is the latest assessment of global mangrove deforestation.

L59: Remove ‘The’

The word is removed.

L60: ‘As a result’ – delete. It’s not necessary, as you set it up in the previous sentence

The statement is removed.

L61: ‘many efforts’ – sounds vague.

The statement is revised.

L66: what’s hydrological rehabilitation?

An explanation of hydrological rehabilitation is added in the following sentence.

L67: ‘Other integrated approaches are increasingly being promoted’. This is vague. Not sure what point is being made here. The ends of the paragraphs should lead the reader on to the next. I’m at the end of the second paragraph now, but still don’t get a sense of what the big question is, what’s important?

The statement is revised.

L71: ‘nursery habitat’ – that’s not a ‘process’. Change to ‘habitat provision’?

The term is revised.

L72: here you state that ‘Similarly, studies have quantified the economic costs and benefits’, but in the first sentence of this paragraph you have already stated that ‘field studies have assessed the ecological and/or economic outcomes...’

The statement is removed.

L80: “adopts a meta-analysis approach” →uses meta-analysis? It’s a method not an approach?

The statement is revised.

L80: “assess” →quantify

The statement is revised.

L81: long sentence, can be cut down considerably by just saying what you quantified L82: ‘various’ – avoid this word. It sounds vague. A range of?

The statement is revised.

Methods

L419: I think you should have searched for mangrove*, rather than mangrove, crab* rather than crab. There are several examples of this.

Thanks for pointing. We adopted the indicated keywords for searching, conducted the literature search again. We found that the results of literature search remain the same.

L438: “valuations from the “grey literature” and documents not in English were not covered in this study”. Why not? How much was there? How do you think this impacted your results?

The decision to exclude grey literature rests on our concerns over possible quality issues as the overwhelming majority of these studies are not peer-reviewed, and report the outcomes of restoration projects for which the reporting organizations have received the funding to undertake the restoration projects.

In our opinion the exclusion of grey literature will not affect the meta-analysis of ecological

outcomes, as few such reports delve deep in the assessment of the ecological outcomes of restoration using the types of paired comparisons that are used in our meta-analysis. A more significant fraction of reports might report the economic costs of mangrove restoration projects.

The exclusion of grey literature is rather common in meta-analyses such as the one reported in this study. The exclusion of grey literature was already illustrated and justified on the first paragraph of the Limitations. We believe that as this is a methodological issue it is more logical to be covered in the Methods rather than the Discussion.

We should also note that some previous studies have tested and suggested that there is no major difference in effect sizes reported in published and unpublished studies in ecological meta-analysis (Møller et al 2005).

L439: ‘In the end’. After what? There has been no description of a robust systematic review protocol, e.g. inclusion/exclusion criteria. Was there any critical appraisal?

We are a bit confused with this comment, as this information about inclusion criteria was originally included in Supplementary Table 1 (first draft of Supplementary Information). Also Supplementary Figure 2 (first draft of Supplementary Information) shows the full flowchart of identified, excluded and included studies across the different databases. Is there a change that the Reviewer did not see this information, or is there something specific that we missed ourselves and we should add?

Clarity on the matter will be very welcome and we believe we have followed and documented all standard information following the PRISMA protocol for meta-analyses and systematic reviews.

We acknowledge that the appraisal of the quality of individual studies was rather informal. Following our response in a previous comment we now formalise the quality appraisal of the individual studies following the criteria of Mupepele et al (2016). We include the outcomes of this assessment as a supporting file for the benefit of the reviewer.

L446: “about 26 types of functions using different variables”. Vague. How were these classified?

The classification and variables was included in the Supplementary Table 2 (first draft of Supplementary Information), cited in this exact sentence. We now expand this table to add the explanation (and relevant citation) and evaluation method for each ecological functions (Supplementary Table 1, revised version of Supplementary Information).

L447: In total 89 studies (representing 975 observations of different functions) met our criteria for inclusion in the meta-analysis. What criteria? Why is this in your section ‘Meta- analysis of restoration outcomes’. It should go in literature selection.

First of all the criteria were included in Supplementary Table 1 in Supplementary Information (first draft). We now move them in main text as we have a feeling based on some previous comments that the Reviewer might not have seen them.

Second, we believe that this is the right point to mention the study selection for the meta-analysis. The previous section identifies the full literature search mentioning the keywords (i.e. pool of studies). This section focuses on the meta-analysis, which make it more logical to mention this selected sub-set of studies here.

L449: “namely”. Unnecessary – you name three?

The statement is revised.

L450-452: This is too late to be describing the comparisons you set out to make. They need to be clearly justified.

The statement is revised. We include more information about the study comparisons.

L463: ‘which provides conventional statistical significance, but also information that cannot be obtained from p values’ I don’t follow. What ‘information’?

The statement is revised.

L465: So you also calculated lnR as well as Hedges’ g? How did you get to the mean percentage of change from lnR? Why did you calculate both effect sizes?

The reviewer asked this as a Major Comment above. We answer fully this comment earlier in that response.

L477: species – not everyone will know what taxa ‘Rhizophora apiculata’ is. Make clear it’s the tree species.

The statement is revised.

L447: plantation; habitat rehabilitation – describe the difference. Where does natural regen fit? You use that term in your intro.

The statement is revised.

L479: ‘origin’ sounds more like the restoration method. I think you mean ‘nativeness’. How did you classify whether species were native or not? What resource was used?

The word is revised. The nativeness of species were illustrated directly in the studies but not determined by the authors.

L485: ‘Elements of the adopted methodology’ – vague. What elements? What is the point of this sentence? Perhaps a bit army-wavey. In general, avoid single-sentence paragraphs.

The statement is revised.

L488: ‘To test the robustness of the results’ □ vague. Robustness in relation to what? Influential studies?

The statement is revised.

L497: ‘met in 30 cases’. What is a ‘case’ here? An article? A study? A comparison? Please define.

The statement is revised. A case refers to an observation.

Results

‘General literature patterns:’ is a little vague for a subheading. Perhaps subheadings could be short statements about the results. Same for other subheadings.

We believe these sub-headings are quite descriptive. Each sub-section has many more results, so we cannot add a very simplified version of the results without creating confusion to reader.

L92 ‘88.8% of reviewed studies’ say how many first. Also, ‘reviewed’ is vague – do you mean those that satisfied review criteria?

The statement is revised.

L102: ‘26 functions,’ - we need a lot more explanation here.

The explanations of ecological functions are added in one column in Supplementary Table 2 (now it is Supplementary Table 1).

Reviewer #3 (Remarks to the Author):

Su et al. conducted a meta-analysis on ecological and economic effects of mangrove restoration. The topic is clearly important and timely. As my expertise is in meta-analysis rather than mangroves, I largely focussed my review on the methods used in this meta-analysis. My specific comments are listed below.

Thanks for the positive feedback. We took on board all your recommendations. We hope that this improved considerably the study.

Methods, line 315. Please specify which databases were used in Web of Science, which is not a database itself but a platform which allows access to a range of databases, and that range depends on institutional subscription. Therefore, your search is not repeatable unless you specify which databases within WoS (e.g. Core Collection) you had access to.

We used the database of WoS core collection. This information is now added in the Methods.

Please use past tense throughout the methods section, currently it is a mixture of past and present (e.g. line 459)

We now use consistently past tense throughout the Methods section.

You say that you used standardized mean difference (Hedges' d) as a measure of effect size, but then on lines 465-466 you say that you calculated mean percentage of change from response ratios. Why then not use response ratios as your effect size measure, i.e. why use two different effect sizes?

In the original manuscript, we used Hedge's g as the effect size metric, which has been widely used in ecological meta-analyses. The mean response ratio ($\ln RR'$) was also calculated in order to estimate the mean percentage of change through $(e^{RR'} - 1) \times 100$. However, this information was not correctly indicated and calculated in the first version of manuscript.

In the revised manuscript, we recalculate the effect size using Response Ratio instead of Hedges' d to avoid confusion. The use of response ratio is quite justified considering its use in nearly half of the meta-analyses in the field of plant ecology (Koricheva et al 2014). We add sentences justifying this methodological selection in the Methods.

Lines 476-483. You say that you conducted subgroup analyses to check for effects of restoration age, but then you also say that you conducted meta-regression to look at restoration age effects. Why do both analyses? Using restoration age as a continuous variable in meta-regression is more powerful approach statistically than comparing effects between studies categorized into different restoration age classes.

In the revised manuscript, we now only estimate the effect of age on effect size through multivariate meta-regression. We delete the estimation of subgroup analysis using age groups.

I was not clear how or whether in your analysis you took into account non-independence between multiple observations from the same study. For instance, presumably in many studies effects of restoration on particular process/function were assessed in multiple years so that multiple effect sizes for that response variable could be derived from the same study. Such repeated measures from the same site/study are not statistically independent and you need to control for this in the analysis, e.g. by including study as a random factor in your model. I could not find any mention in the methods of how or whether you dealt with this problem.

The Reviewer is correct that since there are more than one observation captured in some studies, the issue of non-independence might be considerable in our dataset. In the first draft of the manuscript indeed we did not consider this, which possibly led to inaccurate results and erroneous conclusions.

Following a similar comment by Reviewer 2, in this revised manuscript we use multivariate models instead of standard meta-analytical models (fixed-effect model or random-effect model) to calculate the pooled effect sizes, so that the non-independence in the observed/true effects or outcomes is accounted.

Overall this change in analytical procedure corrects the results generated through the conventional meta-analytical models (random-effect model) used in the first draft. This revision increases the robustness of the meta-analysis as the results of the publication bias tests and the sensitivity analyses indicate that the results of the multivariate models are much more robust compared to the random-effect model used in the previous version.

Supplementary Table 2. You categorized restoration outcomes into broad function type categories (biogeochemical, ecological, anthropocentric). Depending on the specific function measured, the direction/sign of the effect could be interpreted differently. For instance, for biogeochemical functions, higher biomass and carbon storage in restored mangroves is good, but higher emissions of GHG is bad. Similarly, for ecological functions, higher avian diversity in restored mangroves is good, but higher number of invaded species is bad. Did you invert the sign of the effect size for such 'negative' functions before combining them with other functions to calculate the overall effect size? This needs to be explained.

The Reviewer is correct about the “negative” effects, and we appreciate for pointing it out. Following a similar comment by Reviewer 2 we now invert the sign of response ratio for these variables, before combining them to calculate the overall effect size. We indicate this in 1-2 sentences in Methods.

As part of sensitivity analysis you have performed leave-one-out meta-analysis (Supplementary figure 16). It is not clear from this figure and figure caption what is Y axis on all the graphs and what red, blue and black line represent. Instead of leave-one-out analysis, I would prefer to see analysis of effects of outliers (studies with particularly large negative or positive effects) on the results.

Following the Reviewer’s recommendation we now use Cook’s Distance to identify outliers and recalculate the pooled effect size excluding the outlier studies to test the sensitivity of our results.

Overall this analytical change clearly identified the observations of outliers and the effect of outliers on our results. The results of the sensitivity analysis show that 90% of the adjusted estimates remain the same direction and magnitude, indicating the robustness of the results (Supplementary Table 6, Supplementary Figure 8, revised version).

Publication bias and interpretation of trim and fill results. You state (lines 209-210) that results of trim and fill analysis indicate that publication bias might exist. This is incorrect. Trim and fill procedure does not test whether publication bias exists or not, rather, it tests how much your results would change IF publication bias exists. Results of trim and fill procedure that you present in Supplementary Table 6 worry me a great deal because they show that your main results would be reversed if publication bias exists as the direction and/or statistical significance of your main effects changes dramatically when you adjust for possible publication bias. Given that many of your funnel plots are very asymmetrical and suggest that publication bias is likely, it basically suggests that the main results that you are reporting are not robust to publication bias and likely to be overturned if grey literature or unpublished studies were included in the analysis. This is a big concern and needs to be discussed more thoroughly.

Thanks for pointing out our errors in the interpretation of the outcomes of the publication bias analysis. Based on the Reviewer's comment we revise thoroughly both the analysis and the test.

In terms of analysis, the publication bias seems to reduce after using the multivariate models to estimate the effect sizes. For the aggregated results, there is no significant asymmetry in the funnel plots, which means the absence of publication bias and suggests that the robustness of the pooled effect size results. However, there is publication bias for some of the individual functions, but this is in 12 out of 55 studied functions across the different comparisons (Supplementary Figure 10, revised version). To make this more visible we indicate in Figure 1 the functions for which we detected publication bias, and further discuss the issue in the future research part of the Discussion.

In terms of narrative, we revise thoroughly the relevant text, basing it on the interpretation above. We hope we have now avoided errors. We will be happy to revise further if needed.

Geographic bias. One important role of meta-analysis is to highlight current gaps in our knowledge on the topic. Looking at Supplementary figure 3 and comparing mangrove distribution with locations of studies included in your meta-analysis, it is clear that there is a strong geographic bias with lots of studies from Asia but e.g. no studies from Australia. This needs to be acknowledged and discussed in your paper. Is this because mangroves in Australia have not been degraded as much as in Asia and America and hence restoration efforts have not been as widespread? Is there a danger of extrapolating results from your analysis based largely on studies from Asia and America to other regions? Have you tried to account for regional differences in your analysis?

There is certainly literature reporting mangrove restoration in Australia. However, there are no studies assessing the effect of mangrove restoration in Australia meeting the inclusion criteria set for this meta-analysis. It is not possible to ascertain why this happens so we cannot comment.

In any case we do not believe we extrapolate our findings for the entire world. We think we use very careful language, but we will be happy to revise further if needed. To avoid any confusion we add an acknowledgement in the Discussion about this matter.

You have found evidence of increase in magnitude of the effect size with publication year in

some categories of studies (Supplementary figure 17) with reversal in the direction of the effect occurring in a matter of decade. This is very striking, yet you do not really discuss any potential causes of this increase in the ms. Could it be an artefact of more recent studies being conducted in different regions or using different restoration methods?

The issue of the correlation between publication year and effect size disappears after adopting the multivariate meta-regression (Supplementary Figure 9, revised version). Results show that there is no significant correlation between the reported pooled effect size for overall ecosystem outcomes and the publication year in all comparisons. That said, we do not believe we now need to reflect on the above in the manuscript.

Main results. You state in the abstract: "We find that restored mangroves provide many individual functions at levels higher or on par with natural mangroves, naturally-regenerated mangroves, degraded mangroves and unvegetated tidal flats", but Figure 1 tells a very different story. You acknowledge later in the abstract that restoration outcomes vary depending on measured functions and comparative bases, but this needs to be explained more carefully, e.g. for most biogeochemical functions restored mangroves perform much worse than natural mangroves or naturally regenerated mangroves.

The abstract was revised to be clearer and less confusing. We have to note that due to the strict word count we cannot include a more extended version of the results in the abstract.

References

- Koricheva, J. & Gurevitch, J. Uses and misuses of meta-analysis in plant ecology. *J. Ecol.* 102, 828–844 (2014).
- Møller, A. P., Thornhill, R. & Gangestad, S. W. Direct and indirect tests for publication bias: asymmetry and sexual selection. *Anim. Behav.* 70, 497–506 (2005).
- Mupepele, A.-C., Walsh, J. C., Sutherland, W. J. & Dormann, C. F. An evidence assessment tool for ecosystem services and conservation studies. *Ecol. Appl.* 26, 1295–1301 (2016).
- Nakagawa, S., Noble, D. W. A., Senior, A. M. & Lagisz, M. Meta-evaluation of meta-analysis: ten appraisal questions for biologists. *BMC Biol.* 15, 18 (2017).

Reviewers' Comments:

Reviewer #1:

Remarks to the Author:

Mangrove restoration has been receiving a growing attention. With the efforts from mangrove scientists all over the world, the values of this ecosystem have been recognized and reached a consensus (Lee et al., 2019; Friess et al., 2019; 2020). This study emphasized on this important issue by conducting a meta-analysis of peer-reviewed literature on the outcomes of mangrove restoration. The information was well categorized and analyzed.

The followings are my major comments.

Firstly, the abstract, results and the conclusions showed less informative than expected. For example, "The restored mangroves provide significantly higher ecosystem functions than unvegetated tidal flats, lower than natural mangrove stands, and perform on par with naturally-regenerated mangroves and degraded mangroves". This was more like a common sense than a conclusion. If more information (e.g. the values for the comparison, the contributes to the benefit-cost ratios), can be included in results and conclusion, it would be better.

secondly, I was hard to judge the values from Table 2, about combine the restoration with economy. I am not familiar with economy analysis. But I am sure that a newly restored mangrove forest would provide lower services than they are mature. For example, it may take about 10 years for they to become mature and sequester C. It would be the same as other services, coastal protection, timber production, and so on. So, how much would cost for the maintaining and management during these years after restoring and before they had functions? Did you count the values with the time? And how long should be set for these values with the variations among forest types, ages, regions and restoration methods? The instinct values of mangrove forests are more important than the economic benefits. People may restore mangrove forests due to their ecological functions, and base on regional sustainable development, but not for economy.

Thirdly, I have different view of on the implications on the blue economy. The implication of Blue Economy seemed a little far from the above analysis. We know that mangrove restoration can sequestration C and conserve biodiversity, etc., which are the ecosystem functions or services. When comparing to the economy, its contribution to local economy would be lower than maintaining shrimp productions. In the mangrove restoration actions, the replanting area was always limited globally, which may take off a lot of aquaculture ponds and then depress the local economic in a short time. There still need some efforts on changing the economy type, such as some types of sustainable development (e.g. the combination of blue carbon credit, sustainable aquaculture, organic shrimp products, etc. which has been published in previous studies). So, the combination of restoring mangrove with Blue Economy here as the part of "Policy implications and future research" made me read like a jump. I would suggest to focus on the ecological function of restoring mangrove based the meta-analysis for the policy implication.

Reviewer #4:

Remarks to the Author:

In my opinion, this is a strong manuscript for publication. I am not an expert on mangroves but I believe that 1) the premise of this manuscript is sufficiently general to be of interest to a broad array of readers, 2) the manuscript was well written (comments from prior rounds of reviews really helped here) and 3) I thought the statistical analyses were sufficiently justified and interpreted. My only quibble with the manuscript is that it utilizes confidence intervals as the basis for hypothesis tests. This throws a new "critique" on this manuscript but I do not believe it is sufficient to hold this manuscript up. My primary concern with the use of confidence intervals is that it requires one to be tied to a threshold probability level of 5% to assess statistical significance. This is a hotly debated topic but, after looking at the graphs and interpretation of results, I do not believe that readers are being led astray and see merit to the conclusions reached. Reviewer 3 in the prior round of reviews was very insightful and I believe that the

authors did a very good job of addressing the comments raised by this reviewer.

Response to reviewers

Reviewer #1 (Remarks to the Author):

Mangrove restoration has been receiving a growing attention. With the efforts from mangrove scientists all over the world, the values of this ecosystem have been recognized and reached a consensus (Lee et al., 2019; Friess et al., 2019; 2020). This study emphasized on this important issue by conducting a meta-analysis of peer-reviewed literature on the outcomes of mangrove restoration. The information was well categorized and analyzed.

The followings are my major comments.

Firstly, the abstract, results and the conclusions showed less informative than expected. For example, “The restored mangroves provide significantly higher ecosystem functions than unvegetated tidal flats, lower than natural mangrove stands, and perform on par with naturally-regenerated mangroves and degraded mangroves”. This was more like a common sense than a conclusion. If more information (e.g. the values for the comparison, the contributes to the benefit-cost ratios), can be included in results and conclusion, it would be better.

Thank you for your comment. We revised the abstract to include relevant figures for the comparisons, and the results of benefit-cost ratios. However we should point that abstracts are only 150 words so we cannot expand the results in great length.

Originally we avoided adding quantitative estimates in the narrative of the Results and Discussion to improve readability, as all the quantitative estimates were visible in the Figures and Tables. However, following the suggestion of the Reviewer we now add many quantitative estimates throughout the Abstract, Results and Discussion. This reflects also the comment of Reviewer 2 about the 95% CIs.

secondly, I was hard to judge the values from Table 2, about combine the restoration with economy. I am not familiar with economy analysis. But I am sure that a newly restored mangrove forest would provide lower services than they are mature. For example, it may take about 10 years for they to become mature and sequester C. It would be the same as other services, coastal protection, timber production, and so on. So, how much would cost for the maintaining and management during these years after restoring and before they had functions? Did you count the values with the time? And how long should be set for these values with the variations among forest types, ages, regions and restoration methods? The instinct values of mangrove forests are more important than the economic benefits. People may restore mangrove forests due to their ecological functions, and base on regional sustainable development, but not for economy.

The Reviewer is right that such economic benefits would depend on the maturation period of restored mangroves, as this essentially affects the delivery of ecosystem services. We have been aware of this and for this reason in our cost-benefit analysis, we estimated a period of 20 years at different social discount rates (Method section, Line 699 to 700). The economic benefits of restored mangroves were estimated starting in Year 5 (Method section, Line 702) to account for this maturation period. We had also already added a discussion point about the low economic benefit due to the immaturity (Discussion, Line 407-413).

For maintenance and management cost, we calculated annual management costs, which are 2.5% of the capital costs¹ (Method section, Line 705-706) and estimated the total management cost for 20 years under the same discount rates (Equation 1). Unfortunately, for the economic analysis, we are unable to account for the variations among tree species, ages, and restoration methods due to the lack of sufficient information in the source documents, but we used currency exchange calculation to adjust and standardize values from different countries.

In the revised manuscript, we added one sentence in the Discussion (Line 413-415) to declare this limitation. To be even more open about the values used in the economic analysis we added the details of all individual

economic studies we used in one new table in the Supplementary Material (Supplementary Table 7).

Thirdly, I have different view of on the implications on the blue economy. The implication of Blue Economy seemed a little far from the above analysis. We know that mangrove restoration can sequestration C and conserve biodiversity, etc., which are the ecosystem functions or services. When comparing to the economy, its contribution to local economy would be lower than maintaining shrimp productions. In the mangrove restoration actions, the replanting area was always limited globally, which may take off a lot of aquaculture ponds and then depress the local economic in a short time. There still need some efforts on changing the economy type, such as some types of sustainable development (e.g. the combination of blue carbon credit, sustainable aquaculture, organic shrimp products, etc. which has been published in previous studies). So, the combination of restoring mangrove with Blue Economy here as the part of “Policy implications and future research” made me read like a jump. I would suggest to focus on the ecological function of restoring mangrove based the meta-analysis for the policy implication.

We feel that this point is a bit beyond the focus of our analysis. However, it contains some interesting points. Certainly, mangroves and their ecosystem services are being discussed in broader Blue Economy discussions by the World Bank and regional groups such as PEMSEA, CSIRO and others. We now reflect all of the things mentioned by the Reviewer in a new closing paragraph for the Discussion section (Lines 483 to 495).

Reviewer #4 (Remarks to the Author):

In my opinion, this is a strong manuscript for publication. I am not an expert on mangroves but I believe that 1) the premise of this manuscript is sufficiently general to be of interest to a broad array of readers, 2) the manuscript was well written (comments from prior rounds of reviews really helped here) and 3) I thought the statistical analyses were sufficiently justified and interpreted. My only quibble with the manuscript is that it utilizes confidence intervals as the basis for hypothesis tests. This throws a new "critique" on this manuscript but I do not believe it is sufficient to hold this manuscript up. My primary concern with the use of confidence intervals is that it requires one to be tied to a threshold probability level of 5% to assess statistical significance. This is a hotly debated topic but, after looking at the graphs and interpretation of results, I do not believe that readers are being led astray and see merit to the conclusions reached. Reviewer 3 in the prior round of reviews was very insightful and I believe that the authors did a very good job of addressing the comments raised by this reviewer.

Thank you for your critical comment about Confidence Intervals. Traditionally, if the confidence interval of the combined effect size does not include zero, (i.e. in case of a confidence level of 95% the p -value is smaller than 0.05) this means that the meta-analytic effect is statistically significant². However, we aware that this interpretation is problematic and the misuse of p values or CIs as a dichotomous description of “statistical significance” has been criticized^{3,4}.

In the revised manuscript, we modified the explanation about Confidence Intervals in the Methods (Line 651-659) and the footnotes of Figures 2 and 3. In the narrative of the Results we also avoid using the word “significant” when using 95% CIs to discuss the effect of restoration for the delivery of different functions. We hope that the above make even clearer what these results and Figures imply, and thus help the reader appreciate better the results.

References

1. De Groot, R. S. *et al.* Benefits of Investing in Ecosystem Restoration: Investing in Ecosystem Restoration. *Conserv. Biol.* **27**, 1286–1293 (2013).
2. Akobeng, A. K. Understanding systematic reviews and meta-analysis. *Arch. Dis. Child.* **90**, 845–848 (2005).
3. Greenland, S. *et al.* Statistical tests, P values, confidence intervals, and power: a guide to misinterpretations. *Eur. J. Epidemiol.* **31**, 337–350 (2016).
4. Hoekstra, R., Finch, S., Kiers, H. A. L. & Johnson, A. Probability as certainty: Dichotomous thinking and the misuse of p values. *Psychon. Bull. Rev.* **13**, 1033–1037 (2006).

Reviewers' Comments:

Reviewer #5:

Remarks to the Author:

I have only a small number of minor editorial suggestions that I have marked directly on the manuscript.

**Ecological and economic outcomes of mangrove restoration: a meta-analysis**
**approach**

**Jie Su** ^{1,*}, **Daniel A. Friess** ^{2,3}, **Alexandros Gasparatos** ^{4,5}

1 - Graduate Program in Sustainability Science - Global Leadership Initiative (GPSS-GLI), Graduate
School of Frontier Sciences, The University of Tokyo, 5-1-5 Kashiwanoha, Kashiwa City, 277- 8563,
Japan

2 - Department of Geography, National University of Singapore, 1 Arts Link, Singapore 117570

3 - Centre for Nature-based Climate Solutions, National University of Singapore, 16 Science Drive 4,
Singapore 117558

4 - Institute for Future Initiatives (IFI), The University of Tokyo, 7-3-1 Hongo, Bunkyo-ku, Tokyo,
113-8654, Japan

5 - Institute for the Advanced Study of Sustainability (UNU-IAS), United Nations University, 5-53-
70 Jingumae, Shibuya-ku, Tokyo 150-8925, Japan

***Corresponding author. Email address: jie.su@s.k.u-tokyo.ac.jp**

Ecological and economic outcomes of mangrove restoration: a meta-analysis approach

Jie Su, Daniel A. Friess, Alexandros Gasparatos

Keywords

Cost-benefit analysis; Ecosystem services; Mangrove rehabilitation; Ecological restoration

[revised manuscript text omitted]
'=0.64$, $95\%CI=0.35$ to 0.94), nitrogen accumulation ($RR'=0.45$, $95\%CI=0.19$ to 0.72), wastewater treatment ($RR'=0.28$, $95\%CI=0.26$ to 0.29) and wave dissipation ($RR'=1.39$, $95\%CI=0.75$ to 2.02). However, this is not evident for all individual functions, as for example there is no major difference for phosphorus accumulation ($RR'=0.06$, $95\%CI=-0.08$ to 0.21), crab production and diversity ($RR'=1.03$, $95\%CI=-0.64$ to 2.70), and fish production and diversity ($RR'=0.33$, $95\%CI=-0.82$ to 1.84), among others.

156 There is a smaller volume of literature comparing the outcomes of restored mangroves with naturally-
157 regenerated or degraded mangroves in the same studies for the same site. Still, the meta-analysis
158 indicates that restored mangroves have similar levels of restoration outcomes compared to naturally-
159 regenerated mangroves with the same age, and degraded mangroves ($RR'=-0.58$, $95\%CI=-2.25$ to 1.09 ,
160 $RR'=0.13$, $95\%CI=-0.72$ to 0.97 , respectively) (Fig. 2c and 2d). When compared to naturally-
161 regenerated mangroves, only the level of heavy metal accumulation was lower in restored mangroves
162 ($RR'=-0.57$, $95\%CI=-0.87$ to -0.27). There was no major difference for other functions such as carbon
163 sequestration ($RR'=-0.16$, $95\%CI=-0.98$ to 0.66) and other nutrient accumulation ($RR'=-0.23$,
164 $95\%CI=-0.80$ to 0.14). Conversely, restored mangroves exhibit higher levels of crab production and
165 diversity compared to both naturally-regenerated mangroves ($RR'=0.79$, $95\%CI=0.38$ to 1.20) and
166 degraded mangroves ($RR'=0.80$, $95\%CI=0.64$ to 0.97). However, as the number of studies containing
167 matched pairs of restored and naturally-regenerated or degraded mangroves was quite limited, the
168 outcomes of our meta-analysis for these comparisons should be interpreted with caution.

169
170 The Cochran's Q statistic test showed significant heterogeneity across the different restoration
171 outcomes (Supplementary Table 5). As outlined in the next sections, subgroup analysis and meta-
172 regression were conducted to identify the potential effect of different factors on the heterogeneity of
173 the pooled effect sizes.

174
175 **Effect of stand age.** The ages of the studied stands ranged between 1-70 years old, with most studies
176 reporting results from mangrove restoration projects between 1-15 years old (78.8% of cases)
177 (Supplementary Fig. 6). The meta-analysis using restoration age as a predictor suggests a significant
178 correlation between overall restoration outcomes and stand age when the restored mangroves are
179 compared to natural mangroves and unvegetated tidal flats (Supplementary Fig. 7). When looking into
180 individual functions, functions such as biomass production and carbon sequestration significantly
181 increase with mangrove age (slope=0.16, $p<0.0001$, slope=0.03, $p<0.0001$, respectively) when
182 comparing with natural mangroves. Similar increasing patterns are also observed for functions such as
183 organic matter accumulation (slope=0.15, $p<0.0001$) and crab production and diversity (slope=0.24,
184 $p<0.0001$), when comparing restored mangroves with unvegetated tidal flats. Conversely when
185 comparing restored mangroves with natural mangroves, we observe a slight but statistically significant
186 decrease for crab production and diversity (slope=-0.03, $p=0.0001$). This indicates that young restored
187 mangroves may provide better nursery and habitat to larger crab populations than older restored
188 mangroves, with crab production and diversity even in mature restored mangroves being lower
189 compared to natural stands.

190
191 **Effect of restoration species.** The subgroup analysis did not suggest that any individual mangrove
tree species has better restoration outcomes compared to other species in monospecific restoration
settings (Fig. 3a) when comparing restored mangroves with paired natural mangroves. Monospecific
restoration with the four most popular mangrove tree species (*R. apiculata*, *R. mucronata*, *A. marina*,
*K. obovata*) generated considerable ecological effects compared to unvegetated tidal flats (Fig. 3a).
The restoration outcomes for individual functions varied between monospecific and mixed-species
restored mangroves when compared to natural mangroves and unvegetated tidal flats (Supplementary

Fig. 8). For example, when compared to natural mangroves, restored mangroves using mixed-species
performed better than monospecific ones for biomass production ($RR'=-0.41$, $95\%CI=-0.90$ to 0.07
for monospecific vs. $RR'=-0.24$, $95\%CI=-0.54$ to 0.06 for mixed-species). However, for some
ecosystem functions, monospecific restored mangroves generated higher outcomes when compared to
natural mangroves ($RR'=0.02$, $95\%CI=-0.20$ to 0.24 vs. $RR'=-0.33$, $95\%CI=-0.61$ to -0.06 for heavy
metal accumulation) and unvegetated tidal flats ($RR'=0.74$, $95\%CI=0.38$ to 1.10 vs. $RR'=0.51$,
$95\%CI=0.13$ to 0.88 for carbon sequestration) (Supplementary Fig. 8). Overall restored mangroves
with mixed-species have better restoration outcomes when compared to naturally-regenerated
mangroves ($RR'=-0.44$, $95\%CI=-0.88$ to 0.00 vs. $RR'=-0.05$, $95\%CI=-0.31$ to 0.22) (Fig. 3a).

**Effect of restoration method.** In most of the reviewed cases restoration was performed through the
conventional planting method (i.e. plantation) (96.2%), with only eleven articles reporting the effects
of mangrove restoration through hydrological rehabilitation (e.g. drain and trench method). The
subgroup analysis suggests that plantation and hydrological rehabilitation have comparable overall
restoration outcomes (Fig. 3b), but we need to note the much lower number of hydrological
rehabilitation studies ($N=5$) compared to studies on restoration through plantation ($N=83$). For instance,
compared to natural mangroves, the overall restoration outcomes of hydrological rehabilitation
outperformed mangroves restored through plantation ($RR'=-0.10$, $95\%CI=-0.60$ to 0.39 for
hydrological rehabilitation vs. $RR'=-0.22$, $95\%CI=-0.36$ to -0.08 for plantation), while the results are
inverse when compared to unvegetated tidal flats ($RR'=0.28$, $95\%CI=-0.51$ to 1.06 vs. $RR'=0.44$,
$95\%CI=0.23$ to 0.66).

**Effect of species origin.** In terms of overall restoration outcomes, the restoration projects that included
the planting of exotic species (10% of observations) performed better than native species (90%) both
when compared to unvegetated tidal flats ($RR'=0.62$, $95\%CI=0.23$ to 1.01 for using exotic species vs.
$RR'=0.38$, $95\%CI=0.16$ to 0.59 using native species) and natural mangroves ($RR'=-0.21$, $95\%CI=-$
0.39 to -0.02 vs. $RR'=-0.25$, $95\%CI=-0.41$ to -0.09) (Fig. 3c). However, due to limited sample size it
is not possible to estimate the effect size for individual functions for restoration projects that use exotic
species compared with unvegetated tidal flats. When compared to natural mangroves, mangroves
restored with exotic species perform on par with native species for some functions such as phosphorus
accumulation ($RR'=-0.03$, $95\%CI=-0.31$ to 0.25 vs. $RR'=-0.02$, $95\%CI=-0.29$ to 0.26) but lower for
other functions such as carbon sequestration ($RR'=-0.57$, $95\%CI=-0.95$ to -0.18 vs. $RR'=-0.32$,
$95\%CI=-0.54$ to -0.09) (Supplementary Fig. 9).

**Regional variation.** Most of the studies assessing mangrove restoration outcomes focused in
Southeast and East Asia, namely 62.5% of the observations for comparisons with natural mangroves
and 78.1% of the observations for comparisons with unvegetated tidal flats. The results of the subgroup
analysis for comparisons between restored mangroves and natural mangroves do not indicate that
restored mangroves perform better in any one specific region (Fig. 3d). However, when compared to
unvegetated tidal flats, the overall restoration outcomes of restored mangroves were positive in East
Asia ($RR'=0.35$, $95\%CI=0.14$ to 0.55), South America ($RR'=1.89$, $95\%CI=1.46$ to 2.32), and North

America ($RR'=1.69$, 95%CI = 0.60 to 2.78), while in South Asia there was no clear pattern ($RR'=-$
0.02 , 95%CI=-0.29 to 0.25).

<<Insert Fig. 3>>

**Sensitivity analysis and publication bias.** We identified several observations for which Cook's
Distance was greater than traditional threshold of $4/n$ in some functions, suggesting considered high
influence²² (Supplementary Fig. 10). After excluding these outliers, we find that the effects of these
outliers on the pooled effect size are rather minor with same or similar magnitude and direction of
effect size and its 95%CI, and thus the results are robust (Supplementary Table 6).

Temporal change tests suggested no significant correlation between the reported pooled effect size for
overall restoration outcomes and publication year across all group comparisons (Supplementary Fig.
11). Although the effect to ecological functions (for restored mangroves vs. unvegetated tidal flats)
and biogeochemical functions (for restored vs. naturally-regenerated mangroves) show significant

[revised manuscript text omitted]

**Data availability:**

All data used in this study are available at Figshare (<https://doi.org/10.6084/m9.figshare.12901382>).

The source data for plotting figures and tables can also be archived in the above link, except for
Supplementary Table 5, Table 6, Table 8, and Supplementary Fig. 12, which are directly created using
*R* functions. The study quality assessment table is also available in the above link.

**Code availability:**

The code used in this study for meta-analysis is available at Figshare

(<https://doi.org/10.6084/m9.figshare.12901382>).

**References**

[revised manuscript text omitted]

Reviewer #5 (Remarks to the Author):

I have only a small number of minor editorial suggestions that I have marked directly on the manuscript.

Thanks for the positive feedback. As the comments were on draft, we include them below to facilitate the revision. All revisions were made also considering the Formatting Instructions to Authors.

Line 1 and 33: delete 'approach'

We delete the word 'approach'. We also change a bit the title to avoid punctuation marks as per the guidance in the Formatting Instructions to Authors.

Line 46: 'ecosystem functions' is too vague. Can you be more specific?

We cannot be more specific in this part for two reasons. First, we refer to the aggregate restoration outcomes that encompass many different ecosystem functions, which are also different in each comparison due to the availability of the underlying literature. If we add examples of individual functions in the Abstract it will be quite confusing. Second we cannot expand further the abstract as it now stands at 149 words (max. 150 words) and it is stripped to the bare minimum and has to include some level of quantitative information, as per previous reviewer comments.

Line 83: This sentence is ambiguous. Re-word this sentence.

Sentence re-phrased.

Line 90: why not say, 'This paper conducts a meta-analysis to quantify...'

We revise as follow "*Here we conduct a meta-analysis to quantify...*". This meets both the Reviewer comment and follows the recommendation in the Formatting Instructions to Authors on how the last paragraph of the Introduction should start.

Line 309-318: Nice, clear summary.

Thanks for the positive feedback.

Line 436-438: Rewrite sentence.

Sentence revised.

Line 444: add 'at' after 'looking'

Addition made as suggested.

Line 515 "natural processes": This doesn't make sense. Of course natural mangroves are subject to 'natural processes'. Perhaps delete 'or natural processes' or at least clarify. Perhaps the authors mean severe disturbance here?

We revise the sentences for clarity and give examples. Sentence now reads:

"Natural mangroves are generally undisturbed mangroves, i.e. mangroves that have not been affected by severe disturbances, either related to human activity (e.g. land use change, overexploitation) or natural processes (e.g. hypersalinization)".

Line 723: change "not" to "no"

Change made as suggested.